

# Understanding the Mechanism of Arctic Amplification and Sea Ice Loss

Kwang-Yul Kim[1], Jinju Kim[1], Saerim Yeo[2], Hanna Na[3], Benjamin D. Hamlington[4], and Robert R. Leben[5]

[1]School of Earth and Environmental Sciences, Seoul National University, 1 Gwanak-ro, Gwanak-gu, Seoul 08826, Republic of Korea
[2]APEC Climate Center 1463, Haeundae-gu, Busan 48058, Republic of Korea
[3]Ocean Circulation and Climate Research Center, Korea Institute of Ocean Science and Technology, Ansan, 15627, Republic of Korea
[4]Department of Ocean, Earth and Atmospheric Sciences, 4600 Elkhorn Avenue, Room 406, Old Dominion University, Norfolk, Virginia 23529, USA
[5]Colorado Center for Astrodynamics Research, Department of Aerospace Engineering Sciences, ECNT 320, 431 UCB, University of Colorado, Boulder, Colorado 80309-0431, USA

*Correspondence to*: Kwang-Yul Kim (kwang56@snu.ac.kr)

**Abstract.** Sea ice reduction is accelerating in the Barents and Kara Seas. Several mechanisms are proposed to explain the accelerated loss of polar sea ice, which remains an open question. In the present study, the detailed physical mechanism of sea ice reduction in winter is identified using the daily ERA interim reanalysis data. Downward longwave radiation is an essential element for sea ice reduction, but can only be sustained by excessive upward heat flux from the sea surface exposed to air in the region of sea ice loss. The increased turbulent heat flux is used to increase air temperature and specific humidity in the lower troposphere, which in turn increases downward longwave radiation. This feedback process is clearly observed in the Barents and Kara Seas in the reanalysis data. A quantitative assessment reveals that this feedback process is amplifying at the rate of ~8.9 % every year during 1979-2016. Based on this estimate, sea ice will completely disappear in the Barents and Kara Seas by around 2025. Availability of excessive heat flux is necessary for the maintenance of this feedback process; a similar mechanism of sea ice loss is expected to take place over the sea-ice covered polar region when sea ice is not fully recovered in winter.

## 1 Introduction

Over the past decades, rapidly enhanced atmospheric warming has been observed in the Arctic (Serreze and Francis, 2006; Bekryaev et al., 2010; IPCC, 2013). The accelerated warming is pronounced in the lower troposphere during the cold season (Serreze et al., 2009; Screen and Simmonds, 2010a; Screen et al., 2013). An accompanying drastic reduction of sea ice (Comiso et al., 2008; Comiso, 2012) has profound implications for global climate changes by affecting energy exchange between ocean and atmosphere (Serreze and Barry, 2011), and is often referred to as a key factor for accelerated warming in



the Arctic (Holland and Bitz, 2003; Serreze et al., 2007; Screen and Simmonds, 2010a; Kumar et al., 2010). A particularly significant sea ice reduction can be found over the Barents and Kara seas, which potentially influences cold winter extremes over the Eurasian continent (Petoukhov and Semenov, 2010; Overland et al., 2011; Tang et al., 2013; Cohen et al., 2014; Mori et al., 2014; Kim et al., 2014; Kim and Son, 2016). Physically, sea ice loss involves a positive ice-atmosphere

feedback, which leads to an enhanced warming signal in the Arctic region. This feature is generally referred to as Arctic amplification (Screen and Simmonds, 2010a; Serreze and Barry, 2011) and is expected to persist for at least the next decade (IPCC, 2013; Koenigk et al., 2013; Vihma 2014).

Previous studies have proposed the physical mechanisms of Arctic amplification, which involve the effect of atmospheric heat transport (Graversen et al., 2008), oceanic heat transport (Årthun et al., 2012; Chylek et al., 2009;

Spielhagen et al., 2011; Onarheim et al., 2015), cloud and water vapor changes (Francis and Hunter, 2007; Schweiger et al., 2008; Park et al., 2015a; Park et al., 2015b), and/or diminishing sea ice cover (Serreze et al., 2009; Screen and Simonds, 2010a; Kim et al., 2016). The accurate physical process of the Arctic amplification, however, is subject to debate.

Due to the large seasonal variation of insolation, there exists pronounced seasonality in the air-sea interaction process over the Arctic Ocean. During summer, open water readily absorbs solar radiation, which results in increasing heat

content in the oceanic mixed layer. This represents the so-called albedo feedback (Deser et al., 2000; Serreze et al., 2009; Screen and Simmonds., 2010a; Deser et al., 2010; Serreze and Barry, 2011), meaning that the Arctic Ocean is efficient in absorbing atmospheric heat during summer. After sunsets over the Arctic Ocean, the ice-albedo feedback is suppressed and the primary air-sea interaction mechanism becomes oceanic heat transport (Screen and Simmonds, 2010b). The stored heat in the ocean mixed layer is released back to the colder atmosphere above, which will result in warming of the atmosphere.

The decreased insulation effect (Screen and Simmonds, 2010b) due to the loss of sea ice also promotes further sea ice reduction. Thus, heat transfer between the ocean and atmosphere is generally considered as the fundamental mechanism of Arctic amplification, which is pronounced only during the cold season. On the other hand, increased cloud cover and water vapor (Francis and Hunter, 2007; Schweiger et al., 2008; Graversen and Wang, 2009; Park et al., 2015a; Park et al., 2015b) can also contribute to an increase in downward longwave radiation.

Despite the general consensus that heat transfer between the ocean and atmosphere is a crucial element in the physical mechanism of Arctic amplification and sea ice reduction, a quantitative understanding of individual contributions of heat flux components is still controversial. Further, the role of upward and downward longwave radiations in Arctic amplification is vague and not fully understood. Accurately quantifying the contribution of these different mechanisms, therefore, is required for a complete understanding of the Arctic amplification.

In the present study, a quantitative assessment of energy fluxes involved in the Arctic amplification is investigated in relation to the sea ice reduction over the Barents and Kara Seas. This is an extension of the study by Kim et al. (2016) with a specific goal of delineating the feedback mechanism between sea surface and the atmosphere. In particular, we extract a physically meaningful warming/sea ice reduction signal in the Arctic region and investigate how sea ice loss and individual energy fluxes are linked in a quantitative manner. For this goal, cyclostationary empirical orthogonal function (CSEOF)





analysis is carried out on surface and pressure-level variables derived from the ERA interim daily reanalysis data in winter (Dec. 1-Feb. 28, $d = 90$ days).

## 2 Data and Method of Analysis

### 2.1 Data

ECMWF Reanalysis (ERA) interim daily variables are used from 1979-2016 (Dee et al., 2011). Both surface and pressure-level variables during winter (Dec. 1-Feb. 28) are analyzed over the Arctic region (north of 60° N) to understand the detailed physical mechanism of sea ice melting and Arctic amplification.

### 2.2 CSEOF analysis and regression analysis in CSEOF space

Analysis tool used for this study is the CSEOF technique (Kim et al., 1996; Kim and North, 1997; Kim et al., 2015). In
CSEOF analysis individual physical processes in space-time data are decomposed as:

$$T(r,t) = \sum_n B_n(r,t)T_n(t), \qquad B_n(r,t) = B_n(r,t+d), \tag{1}$$

where $B_n(r,t)$ depicts daily winter evolution of the $n$th physical process and $T_n(t)$ describes how the amplitude of the evolution varies on a longer time scale. Since the nested period $d = 90$ days, each loading vector, $B_n(r,t)$, consists of 90 spatial patterns which depict evolution of a variable throughout the winter. These winter evolution patterns, $B_n(r,t)$, repeat
every winter, but its amplitude varies from one year to another. CSEOF loading vectors are mutually orthogonal to each other in space and time and are deemed to represent distinct physical processes. The principal component (PC) time series, $T_n(t)$ are uncorrelated with (and are often nearly independent of) each other. Thus, the CSEOF technique is suitable for extracting and depicting temporal evolution of (nearly independent) physical processes and often yields valuable insight that cannot be attained from single spatial pattern.

In order to make suitable physical interpretation of the analysis results, CSEOF analysis is conducted on a number of key variables. It is, then, extremely important to make CSEOF loading vectors derived from individual variables to be physically consistent with each other. For the purpose of generating physically consistent CSEOF loading vectors, regression analysis is carried out in CSEOF space (Kim et al., 2015). A target variable is chosen such that its major CSEOF modes best depict the physical processes under investigation; target variable is sea ice concentration in the present study.

Once CSEOF analysis on the "target" variable is completed as in (1), physically consistent loading vectors of another variable, called the "predictor" variable, are obtained as follows:

Step 1: $P(r,t) = \sum_n C_n(r,t)P_n(t)$     (CSEOF analysis on a new variable)     (2)

Step 2: $T_n(t) = \sum_{m=1}^{M} \alpha_m^{(n)} P_m(t)$     (regression on PC time series)     (3)

Step 3: $Z_n(r,t) = \sum_{m=1}^{M} \alpha_m^{(n)} C_m(r,t)$     (regressed loading vector)     (4)

Then, the target and predictor variables can be written as



$$\{T(r,t), P(r,t)\} = \sum_n \{B_n(r,t), Z_n(r,t)\} T_n(t). \tag{5}$$

Namely, the loading vectors of the two variables, $B_n(r,t)$ and $Z_n(r,t)$, share an identical PC time series, $T_n(t)$, for each mode. As a result, the evolution of a physical process manifested as $B_n(r,t)$ and $Z_n(r,t)$ in two different variables is governed by a single amplitude time series. Otherwise, $B_n(r,t)$ and $Z_n(r,t)$ do not represent the same physical process and

henceforth are not physically consistent. This process can be repeated for other predictor variables. As a result of regression, then, entire data can be written in the form

$$Data(r,t) = \sum_n \{B_n(r,t), Z_n(r,t), U_n(r,t), \dots\} T_n(t), \tag{6}$$

where the terms in curly braces denote physically consistent evolutions derived from various physical variables. Aside from the winter seasonal cycle, the first CSEOF mode derived from the daily winter sea ice concentration data in the Arctic

depicts sea ice loss and associated Arctic warming in the Barents and Kara Seas.

**3 Results and Discussion**

 Figure 1 shows the winter-averaged pattern of $B_1(r,t)$ together with the regressed patterns from other variables (the terms in the curly braces in (6)). We refer to it as the sea ice loss mode, since the loading vector (Fig. 1a; see also Fig. 2) and the amplitude time series (Fig. 1g) describes the sea ice reduction, together with natural variability of sea ice concentration, in

the Barents and Kara Seas during the past 37 years (Fig. 1h). In particular, the rate of sea ice loss has significantly increased since 2004-2005 (Vihma, 2014). In association with the sea ice reduction, 2 m air temperature, 850 hPa temperature, specific humidity, upward longwave radiation, downward longwave radiation, and upward heat flux have increased significantly over the region of major sea ice reduction (21°-79.5° E × 75°-79.5° N) (black boxed area in Fig. 1a). As can be seen in Figs. 1a, 1c and 1e, the central areas of anomalous 2 m air temperature, upward longwave radiation and turbulent (sensible + latent)

heat flux match well with the region of sea ice loss (Screen and Simmonds, 2010b). On the other hand, the centers of downward longwave radiation and specific humidity match well with that of the 850 hPa air temperature (Figs. 1b, 1d, and 1f).

 Sea ice concentration varies slightly on a daily basis, and it remains to be nearly stationary throughout the winter (Fig. 2). In accordance with the reduced sea ice concentration, upward longwave radiation flux is increased from the warmer

sea surface exposed to air. Judging from the amplitude time series of the sea ice loss mode (Fig. 1h), sea ice concentration has been reduced by ~40% during the last 35 years.

 Figure 3 shows the anomalous surface (2 m) air temperature and 925 hPa air temperature (upper panels) and the vertical section of anomalous temperature, geopotential height and wind along 60°E and 80°N associated with sea ice reduction. A significant warming is seen in the lower troposphere (e.g., Serreze and Francis, 2006; Serreze et al., 2007;

Screen et al., 2013). Note that the anomalous temperature pattern is similar to the second EOF pattern in Graversen et al. (2008). The anomalous temperature and geopotential height are consistent according to the hydrostatic equation. Anomalous wind and geopotential height are consistent according to the thermal wind equation. As can be seen, an anticyclonic





circulation is established over the region of sea ice loss. This anticyclonic circulation results in advection of warmer air over the Barents and Kara Seas and advection of colder air over the mid-latitude East Asia (Kim and Son, 2016).

The winter-averaged patterns of anomalous downward longwave radiation and specific humidity look fairly similar to that of 850 hPa air temperature (Figs. 4a and 4b). It appears that the increased downward longwave radiation is the result

of the tropospheric warming (Fig. 3). Specific humidity also increases with the tropospheric warming. Note specifically that these changes are observed over or close to the region of sea ice reduction. The pattern of total cloud cover, on the other hand, differs significantly from that of sea ice concentration or downward longwave radiation (Fig. 4d) (Screen and Simmonds, 2010a). The pattern of total cloud cover associated with the sea ice loss mode does not exhibit any strong cloud activity over the region of sea ice reduction, suggesting little connectivity between sea ice reduction and change in cloud

cover. It should be understood, however, that cloud cover is a difficult variable to simulate accurately in a reanalysis model. Therefore, we postulate that the increased downward longwave radiation is due to the increased 850 hPa air temperature and the greenhouse effect produced by the increased specific humidity. Further note that net (upward minus downward) longwave radiation is positive over the region of major sea ice reduction, whereas it is slightly negative over the surrounding areas (Fig. 4c). Thus, at the surface level, there is a net loss of longwave energy over the region of sea ice reduction, while

there is a net gain of longwave radiation over the surrounding area.

A prominent source of energy available for heating the atmospheric column is the increased turbulent heat flux from the sea surface exposed to air due to sea ice reduction (Fig. 5). Although the total (area-weighted) magnitudes of sensible and latent heat fluxes are generally smaller than those of upward and downward longwave radiation (see Fig. 6a), turbulent heat flux (see Fig. 5) is locally more pronounced than longwave radiations (Deser et al., 2010). Furthermore, the combined effect

of turbulent heat flux is about 6 times larger than that of longwave radiation, since upward and downward longwave radiation tends to offset each other and the resulting net longwave radiation is comparatively smaller than the net upward turbulent heat flux (Fig. 6a). In the presence of turbulent heat flux, air temperature and, henceforth, downward longwave radiation can increase continually leading to further sea ice reduction.

While the increased downward longwave radiation is a key element of sea ice reduction, it is not a sustainable

physical process by itself. The area-averaged magnitudes of the upward and downward longwave radiation exceed those of the sensible and latent heat flux in the Barents and Kara Seas (Fig. 6a). The net amount of upward longwave radiation, however, is much smaller than the net upward heat flux as a result of near cancellation between the upward and downward longwave radiations. In fact, the upward radiation is, in general, slightly larger than the downward radiation resulting in the net upward longwave radiation of ~2 W m$^{-2}$ in winter in the Barents and Kara Seas. This implies that surface air temperature

should decrease, preventing further sea ice reduction. A decrease in surface air temperature also means that tropospheric air temperature should decrease. In this sense, downward longwave radiation alone is not sufficient to sustain the sea ice reduction process. On the other hand, the net amount of heat flux is ~12 W m$^{-2}$ in the same area. Once ocean surface is exposed due to the reduction of sea ice by ocean current (Schlichtholz, 2011; Smedsrud et al., 2013) or wind (Park et al., 2015b), the enhanced turbulent heat flux helps sustain sea ice reduction.




As can be seen in Figs. 6b and 6c, daily upward longwave radiation change over the sea ice loss region is highly correlated with the daily fluctuation of 2 m air temperature, whereas daily downward longwave radiation change is strongly correlated with both 850 hPa and 2 m air temperatures. According to the lagged correlations (Fig. 7), daily changes of both upward and downward longwave radiations in the sea ice loss mode are highly correlated with those of 2 m air temperature

and 850 hPa air temperature to a lesser extent. According to analysis based on 3-hourly data, 850 hPa air temperature leads changes in downward longwave radiation. Change in 2 m air temperature, on the other hand, is nearly simultaneous with the downward longwave radiation, whereas it slightly leads the upward longwave radiation. It appears that the increased tropospheric temperature increases the downward longwave radiation. As a result, 2 m air temperature increases and subsequently upward longwave radiation increases. Further, both downward and upward longwave radiation changes seem

to lead sea ice concentration change.

Therefore, we propose a feedback mechanism as suggested in Fig. 8. Sea ice reduction in this area leads to an increase in upward heat flux, which is used to raise temperature in the lower troposphere. Warming in the lower troposphere increases downward longwave radiation. As a result, surface air temperature increases and sea ice melts. This feedback process can be written mathematically as follow:

Step 1: $\frac{dFL^{\uparrow}}{dt} = -\alpha \frac{dS}{dt}$,      $FL^{\uparrow} = SW^{\uparrow} - SW^{\downarrow} + LW^{\uparrow} - LW^{\downarrow} + SF^{\uparrow} + LF^{\uparrow}$,      (7)

Step 2: $\frac{dT}{dt} = \beta \frac{dFL^{\uparrow}}{dt}$,      (8)

Step 3: $\frac{dLW^{\downarrow}}{dt} = \gamma \frac{dT}{dt}$,      (9)

Step 4: $\frac{dS}{dt} = -\delta \frac{dLW^{\downarrow}}{dt}$,      (10)

where $S$ is sea ice concentration, $T$ is tropospheric (850 hPa) temperature, $LW^{\downarrow}$ is downward longwave radiation, and the net

upward flux $FL^{\uparrow}$ is the sum of net short and longwave radiations and sensible and latent heat fluxes. According to the winter (90-day) averaged loading vector of the sea ice loss mode, $\alpha = 1.016 \times 10^{2}$, $\beta = 9.522 \times 10^{-2}$, $\gamma = 1.155 \times 10^{1}$, and $\delta = 8.946 \times 10^{-3}$. It is emphasized that sea ice reduction continues, since downward longwave radiation continues to increase via enhanced upward heat flux from the exposed sea surface. According to our model, 1 % reduction in sea ice coverage leads to 1.02 W m$^{-2}$ increase in upward energy flux, which, in turn, leads to 0.09 K increase in 850 hPa air

temperature and 0.91 W m$^{-2}$ increase in downward longwave radiation. As a result, 2 m air temperature increases by 0.24 K (see Fig. 6).

Note that this feedback mechanism, in its present form, does not require any delayed action of increased absorption of insolation during summer in terms of albedo feedback. In winter, a significant amount of turbulent heat flux can be released from the ocean exposed to cold air without excessive energy stored in summer. Summer heating, on the other hand,

may be a fortifying factor for this feedback loop by preventing sea ice from refreezing during fall and winter.



According to the amplitude time series in Fig. 1g, the rate of sea ice melting appears to be accelerating. A curve fit with an exponential function results in

$$pc(t) = a\exp(\lambda t) + b = a(e^{\lambda})^t + b \triangleq a(1 + \lambda)^t + b, \tag{11}$$

where $pc(t)$ is the amplitude time series in Fig. 1g, and t is time in years since 1979. We obtained the fitting curve (dashed

curve in Fig. 1g) with parameters $a = 1.275 \times 10^{-1}$, $\lambda = 8.916 \times 10^{-2}$, and $b = -9.055 \times 10^{-1}$. Equation (11) can be rewritten as

$$pc(t) - c = (pc(0) - c)(1 + \lambda)^t . \tag{12}$$

That is, the amplitude of sea ice melting and atmospheric warming increases at the rate of ~8.9 % every year.

**4 Concluding Remarks**

A quantitative estimation of changes in the sea ice and other key variables in the Barents and Kara Seas reveals that increase in downward longwave radiation is sustained by an increase in turbulent flux from the exposed sea surface. While a wider area of sea surface is exposed to air and upward longwave radiation increases due to summer sea surface warming, the increased upward longwave radiation alone seems insufficient to produce a feedback loop. Due to a net deficit of surface radiation in fall/winter, sea ice may refreeze quickly (see Figs. 7 and 8 in Kim et al., 2016). Prolonged sea ice melting is

instrumental for increased turbulent flux, which in turn warms the atmospheric column (see Fig. 5). As a result, downward longwave radiation increases and sea ice reduction continues in accordance with surface warming (Fig. 8). This is why significant Arctic amplification is observed only in the Barents and Kara Seas but not in the Laptev, East Siberian or Chukchi Seas, where summer sea ice melting is conspicuous but sea ice quickly refreezes in late fall/early winter (Kim et al., 2016). How sea ice refreezing is delayed in the Barents and Kara Seas remains to be answered. Sea ice cover in the Barents

and Kara Seas was ~80 % in 1979 and is currently ~40 %. Our calculation shows that sea ice in the sea-ice loss region (21°-79.5° E × 75°-79.5° N) of the Barents and Kara Seas may completely melt by around 2025 (Fig. 1h) unless impeded by other naturally occurring variability.

It should be pointed out that this feedback process could develop in other areas of the Arctic Ocean. If sea ice refreezing is delayed in late fall/winter, increased turbulent heat flux from the open sea surface will make it more difficult for

sea surface to refreeze, ultimately leading to the feedback process in Fig. 8. It is, of course, difficult to determine when this should occur, since environmental factors differ from one location to another.

**5 Data and code availability**

All the results of analysis and the programs used in the present paper are freely available by contacting the corresponding author.




*Acknowledgments.* This research was supported by the National Science Foundation of Korea.





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





**Figure Captions**

**Figure 1**. Winter (Dec. 1-Feb. 28) average patterns of sea ice loss mode: (a) sea ice (shading) and 2 m air temperature (contour), (b) 1000-850 hPa specific humidity, (c) upward longwave radiation, (d) downward longwave radiation, (e) turbulent (sensible + latent) heat flux, (f) 850 hPa air temperature, (g) the corresponding amplitude change (red solid curve) and the amplification curve (blue dashed curve), and (h) actual sea ice change in the sea-ice loss region (21°–79.5° E × 75°–79.5° N; the boxed area in (a)) of the Barents and Kara Seas (black dotted curve), sea ice change according to the sea ice loss mode (red curve), projection based on the amplification curve (blue dashed curve). The green and purple contours in (b)-(f) represent sea ice concentration in (a). The numbers in parenthesis are contour intervals and negative contours are dashed.

**Figure 2.** (a) Anomalous daily sea ice concentration and (b) upward longwave radiation averaged over the region of sea ice loss (21°-79.5° E × 75°-79.5° N). Winter days are counted from December 1.

**Figure 3**. Winter-averaged patterns of atmospheric condition: (a) 2m air temperature, (b) 925 hPa air temperature, (c) vertical cross section along 60° E of lower tropospheric (1000-850 hPa) air temperature, geopotential height and wind, and (d) along 80°N. Contour intervals are in parenthesis in (a) and (b). Temperature is in shading (0.4 K), geopotential height is in black contours (3 m), and (c) zonal and (d) meridional winds are in blue contours (0.2 m s$^{-1}$).

**Figure 4**. Winter-averaged patterns of (a) 850 hPa air temperature (shading) and 2 m air temperature (contour), (b) 900-hPa specific humidity (shade) and downward longwave radiation at surface (contour), (c) net (upward minus downward) longwave radiation at surface (shade) and SAT (contour), and (d) total cloud cover for the sea ice loss mode. The green and purple contours in (a)-(d) represent the reduction of sea ice concentration.

**Figure 5**. Winter average pattern of sea ice loss mode in the Barents and Kara Seas: (a) sea ice (%, shading), 2 m air temperature (red contour) and 850 hPa temperature (black contour), (b) upward longwave radiation (red contour) and downward longwave radiation (black contour), (c) sensible heat flux (red contour) and latent heat flux (black contour), and (d) net energy balance (sensible heat flux + latent heat flux + upward longwave radiation – downward longwave radiation).

**Figure 6**. Daily patterns of variability over the region of sea ice loss (21°-79.5° E × 75°-79.5° N): (a) upward longwave radiation (blue dashed), downward longwave radiation (blue dotted), net longwave radiation (blue solid) with its mean value (blue straight line), sensible heat flux (red dashed), latent heat flux (red dotted), and turbulent heat flux (red solid) with its mean value (red straight line), (b) 2 m air temperature (red), 850 hPa air temperature × 2 (black), and upward longwave radiation (blue), and (c) same as (b) except for the regressed downward longwave radiation (blue). The straight lines in (b) and (c) represent the winter mean value of anomalous 2 m air temperature. Correlation of upward and downward longwave radiations with 2 m air temperature is respectively 0.88 and 0.91, whereas with 850 hPa air temperature is 0.66 and 0.85.

**Figure 7.** Lagged correlations: (a) correlation of upward (solid lines) and downward (dotted lines) longwave radiations with 2 m air temperature (blue), 850 hPa temperature (red), and sea ice concentration (black), and (b) a blow-up of the boxed region in (a). Longwave radiation lags the other variable for a positive lag. Lagged correlation between 2 m air temperature and 850 hPa air temperature (black dashed line); 2 m air temperature leads 850 hPa temperature for a positive lag.



**Figure 8**. A proposed mechanism of polar amplification. Increased net upward energy flux increases air temperature. As a result, downward longwave radiation increases, which results in warmer surface temperature and sea ice melting. This loop seems to amplify by ~8.9% annually.





(a) SIC (2 %) & 2m AIR T (0.5° C)

(b) 1000-850 hPa SH (3×10⁻³ g Kg⁻¹)

(c) ULW at SFC (2 W m⁻²)

(d) DLW at SFC (2 W m⁻²)

(e) TURBULENT FLUX (4 W m⁻²)

(f) 850 hPa T (0.2° C)





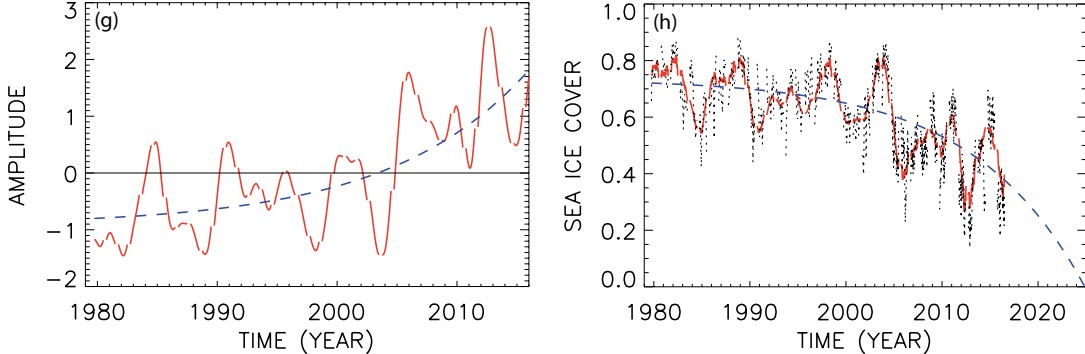

**Figure 1.** Winter (Dec. 1-Feb. 28) average patterns of sea ice loss mode: (a) sea ice (shading) and 2 m air temperature (contour), (b) 1000-850 hPa specific humidity, (c) upward longwave radiation, (d) downward longwave radiation, (e) turbulent (sensible + latent) heat flux, (f) 850 hPa air temperature, (g) the corresponding amplitude change (red solid curve) and the amplification curve (blue dashed curve), and (h) actual sea ice change in the sea-ice loss region (21°–79.5° E × 75°– 79.5° N; the boxed area in (a)) of the Barents and Kara Seas (black dotted curve), sea ice change according to the sea ice loss mode (red curve), projection based on the amplification curve (blue dashed curve). The green and purple contours in (b)-(f) represent sea ice concentration in (a). The numbers in parenthesis are contour intervals and negative contours are dashed.





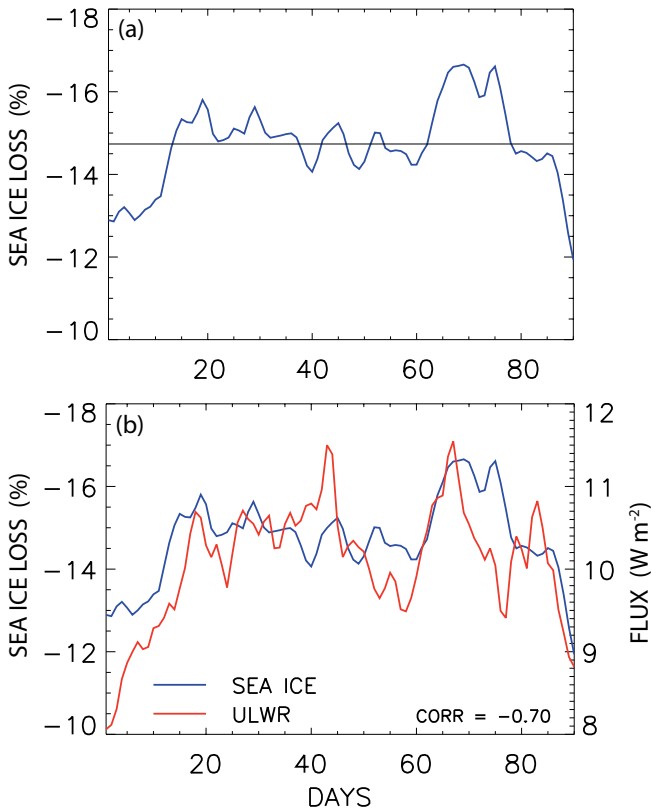

**Figure 2.** (a) Anomalous daily sea ice concentration and (b) upward longwave radiation averaged over the region of sea ice loss (21°-79.5° E × 75°-79.5° N). Winter days are counted from December 1.





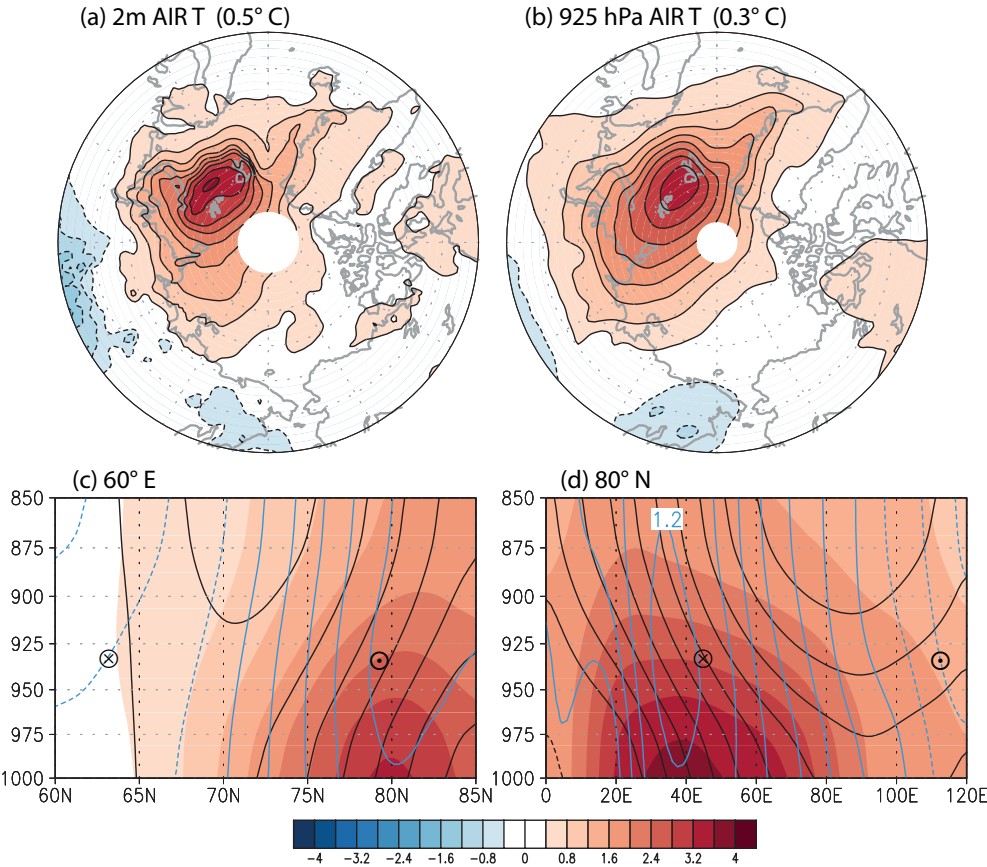

**Figure 3.** Winter-averaged patterns of atmospheric condition: (a) 2 m air temperature, (b) 925 hPa air temperature, (c) vertical cross section along 60° E of lower tropospheric (1000-850 hPa) air temperature, geopotential height and wind, and (d) along 80° N. Contour intervals are in parenthesis in (a) and (b). Temperature is in shading (0.4 K), geopotential height is in black contours (3 m), and (c) zonal and (d) meridional winds are in blue contours (0.2 m s$^{-1}$).



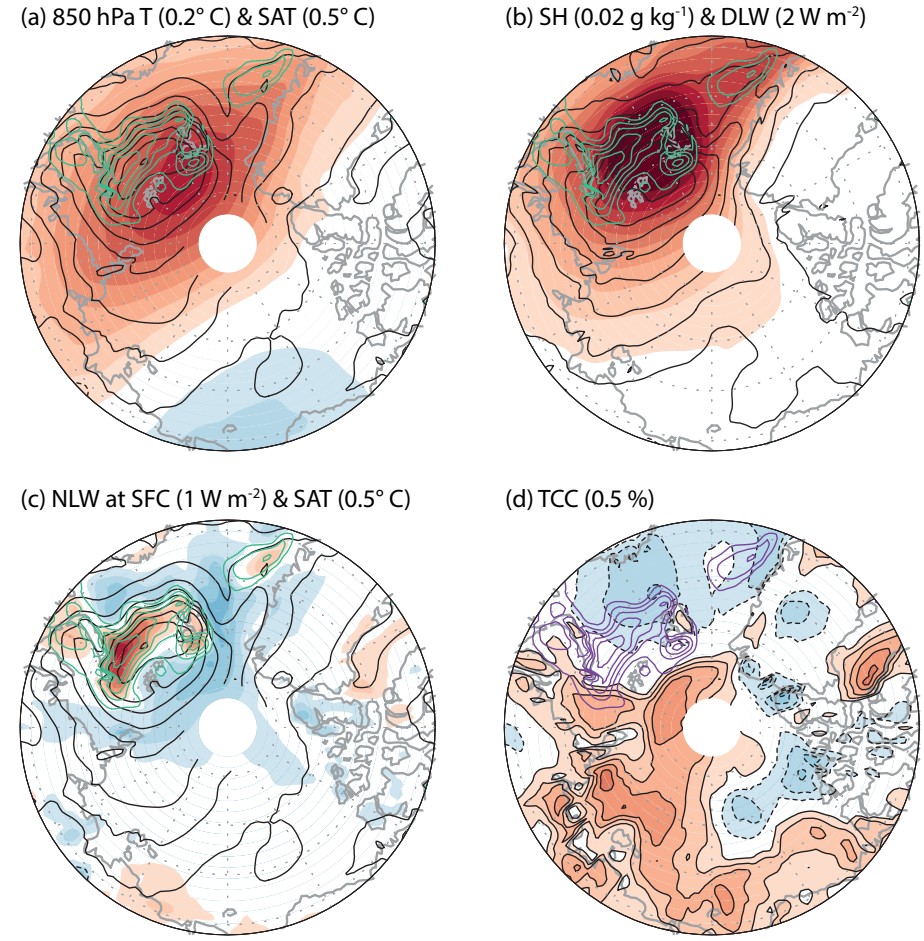

**Figure 4**. Winter-averaged patterns of (a) 850 hPa air temperature (shading) and 2 m air temperature (contour), (b) 900-hPa specific humidity (shade) and downward longwave radiation at surface (contour), (c) net (upward minus downward) longwave radiation at surface (shade) and SAT (contour), and (d) total cloud cover for the sea ice loss mode. The green and purple contours in (a)-(d) represent the reduction of sea ice concentration.





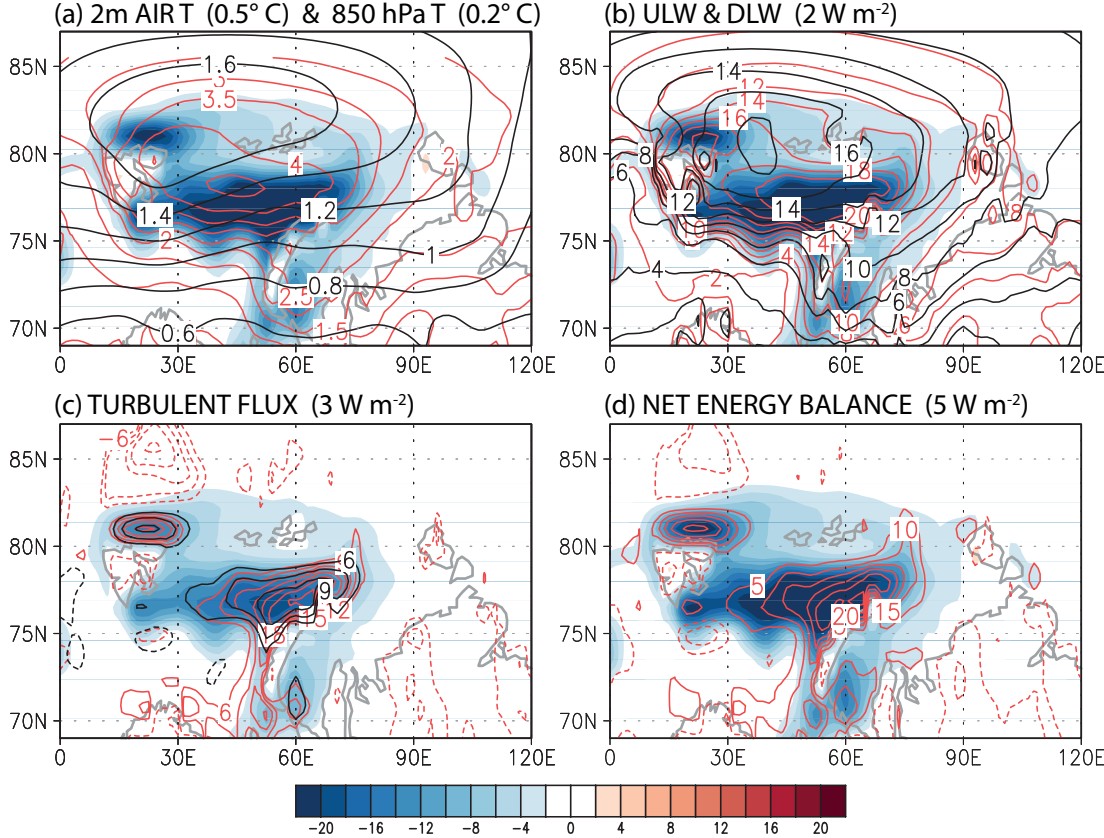

**Figure 5**. Winter average pattern of sea ice loss mode in the Barents and Kara Seas: (a) sea ice (%, shading), 2 m air temperature (red contour) and 850 hPa temperature (black contour), (b) upward longwave radiation (red contour) and downward longwave radiation (black contour), (c) sensible heat flux (red contour) and latent heat flux (black contour), and (d) net energy balance (sensible heat flux + latent heat flux + upward longwave radiation – downward longwave radiation).



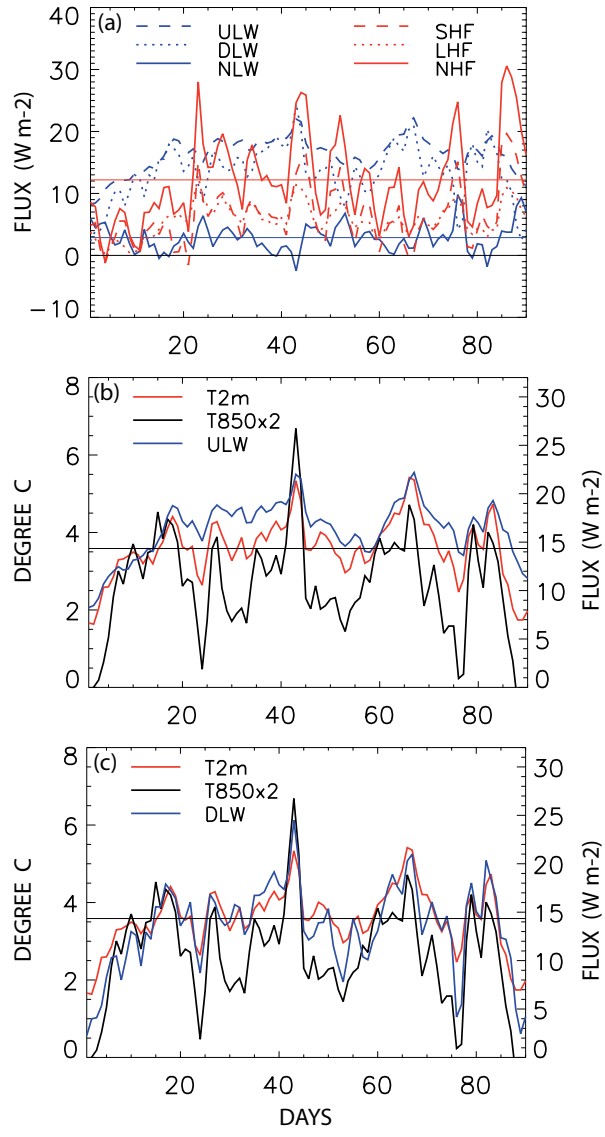

**Figure 6.** Daily patterns of variability over the region of sea ice loss (21°-79.5° E × 75°-79.5° N): (a) upward longwave radiation (blue dashed), downward longwave radiation (blue dotted), net longwave radiation (blue solid) with its mean value (blue straight line), sensible heat flux (red dashed), latent heat flux (red dotted), and turbulent heat flux (red solid) with its mean value (red straight line), (b) 2 m air temperature (red), 850 hPa air temperature × 2 (black), and upward longwave radiation (blue), and (c) same as (b) except for the regressed downward longwave radiation (blue). The straight lines in (b) and (c) represent the winter mean value of anomalous 2 m air temperature. Correlation of upward and downward longwave radiations with 2 m air temperature is respectively 0.88 and 0.91, whereas with 850 hPa air temperature is 0.66 and 0.85.





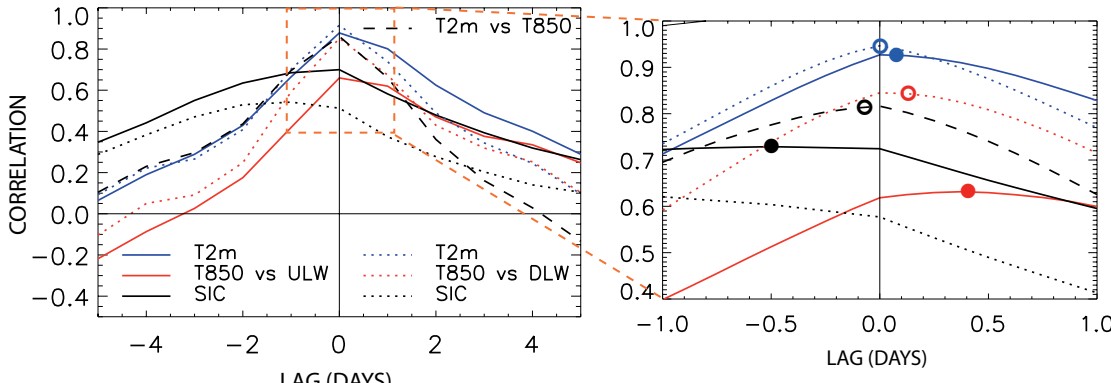

**Figure 7**. Lagged correlations: (a) correlation of upward (solid lines) and downward (dotted lines) longwave radiations with 2 m air temperature (blue), 850 hPa temperature (red), and sea ice concentration (black), and (b) a blowup of the boxed region in (a). Longwave radiation lags the other variable for a positive lag. Lagged correlation between 2 m air temperature and 850 hPa air temperature (black dashed line); 2 m air temperature leads 850 hPa temperature for a positive lag.





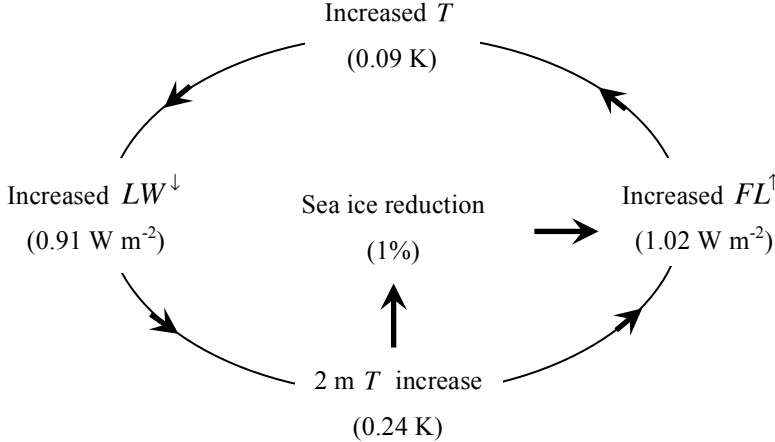

**Figure 8.** A proposed mechanism of polar amplification. Increased net upward energy flux increases air temperature. As a result, downward longwave radiation increases, which results in warmer surface temperature and sea ice melting. This loop seems to amplify by ~8.9 % annually.