# Peer review of "Understanding the Mechanism of Arctic Amplification and Sea Ice Loss"

_The Cryosphere, 2017_

## Referee Comment (RC1) · Anonymous Referee #1 · 11 May 2017

The manuscript attempts to address the mechanisms of Arctic amplification of climate warming and sea ice loss. Cyclostationary empirical orthogonal functions are applied on ERA-Interim reanalysis products, and the methodology includes novel aspects. Some interesting results are found on the relationships between turbulent surface fluxes, longwave radiation, and sea ice loss. After substantial revisions the manuscript has potential for a good paper in The Cryosphere.

Major comments

1. The authors focus on the statistical relationships of the spatial patterns of anomalies in wintertime sea ice concentration, turbulent surface fluxes of sensible and latent heat, upward and downward longwave radiation, as well as air temperature, humidity and total cloud cover. Both the Arctic amplification and sea ice loss are, however,
much more complicated processes, involving many factors, such as the large-scale atmospheric transports of heat and moisture from lower latitudes to the Arctic, and oceanic transports of heat and freshwater. Also, the role of clouds in the Arctic climate system cannot be characterized simply by the total cloud cover; the cloud water and ice contents are at least equally important, as are also the complex interactions between the surface fluxes, boundary-layer turbulence, cloud physics and radiative transfer. I don't mean that the authors should address all these processes, but they should make it clear in the manuscript that they restrict to processes acting in the Arctic, ignoring the forcing from lower latitudes, and they should pay more attention to cloud water and ice contents, which are physically more meaningful variables than the total cloud cover based on reanalysis.

2. Atmospheric reanalyses include serious errors in the Polar regions. This is the case particularly for surface fluxes and near-surface meteorological variables (e.g. Jakobson et al., 2012; Tastula et al., 2013). Hence, to obtain more robust results, I suggest to repeat the calculations using a second reanalysis (e.g. NCEP-CFSR) in addition to ERA-Interim.

3. Several results detected from the reanalysis require a better physical explanation.

(a) Lines 32-33: Why does the region of sea ice loss generates anticyclonic circulation? Figure 3 is not clear in this respect. Show maps of sea level pressure or geopotential height at a relevant pressure level to illustrate this and explain the physical mechanism resulting in an anticyconic circulation.

(b) Page 5, lines 29-30: if the turbulent surface fluxes are upward and net longwave radiation is upwards, they tend to reduce the Earth surface temperature but increase the near-surface air temperature, not decrease it.

(c) Page 6, lines 6-7: Why does a change in 2 m air temperature slightly lead the upward longwave radiation? Further, on lines 8-9: an increase in 2-m air temperature does not have a causal effect of increasing the upward longwave radiation (a statistical

relationship may naturally exist). The upward longwave radiation at the surface (which is the product archived in ERA-Interim) is controlled by the surface temperature (and emissivity), not by the 2-m air temperature. Further, on line 10: Instead of a causal effect from upward longwave radiation to sea ice concentration change, a reduction of sea ice concentration in winter must have an immediate effect in strongly increasing the upward longwave radiation.

(d) Page 6, line 13: related to the above, I suggest removing the words "surface air temperature increases and".

Minor comments

Page 2, line 7: I think Vihma (2014) should be dropped from this line.

Page 2, lines 13-17: During spring and early summer the albedo decreases from roughly 0.85 of dry snow-covered ice to 0.4 of melting ice. Hence, the albedo feedback is important already during the snow and ice melt, already before the appearance of open sea.

Page 2, line 18: the term "oceanic heat transport" is not the best possible, as it may be interpreted as the horizontal transport from lower latitudes to the Arctic. Equation 1 and the text below: explain what is r.

Page 4, line 23: On the basis of Figure 2, I would not write that the sea ice concentration remains nearly stationary throughout the winter.

Page 4, line 25: I cannot detect the 40% decrease from the amplitude time series. Also, better explain what the amplitude represents.

Page 5, line 9: "the little connectivity" between sea ice reduction and total cloud cover may originate from the fact that sea ice reduction generates two effects that compete against each other: increased latent heat flux tends to increase cloudiness but increasing sensible heat flux tends to reduce it.

Page 6, lines 23-25: These are interesting numbers. Please, confirm if these are winter means in 1979-2016 averaged over sea areas north of 60N. It might be interesting to compare them against results of Lupkes et al. (2008), which show air temperature responses to 1% reduction in sea ice cover in different conditions.

Figure 2. Is plot (a) needed at all, as the same line appears in plot (b)?

Figure 4. Add information on colour scales and absolute values. Only the contour intervals are given

Figure 5. Does the shading represent sea ice concentration in all four plots? If yes, why it includes small differences between the plots?

Figure 6. Explain better how the time series in days should be interpreted. It cannot be the mean over 1979-2016. Is it from some selected year?

Figure 8. Referring to my previous comments, I suggest dropping "2m T increase (0.24)" from the figure, and drawing an arrow directly from increased LW-down to sea ice reduction. Otherwise, provide a good explanation on the causality of the link.

References:

Jakobson, E., T. Vihma, T. Palo, L. Jakobson, H. Keernik, and J. Jaagus (2012). Validation of atmospheric reanalyzes over the central Arctic Ocean, Geophys. Res. Lett. 39, L10802, doi:10.1029/2012GL051591.

Tastula, E.-M., T. Vihma, E. L. Andreas, and B. Galperin (2013), Validation of the diurnal cycles in atmospheric reanalyses over Antarctic sea ice, J. Geophys. Res. Atmos., 118, 4194–4204, doi:10.1002/jgrd.50336.

---

## Referee Comment (RC2) · Anonymous Referee #2 · 15 May 2017

The paper applies a sort of regression analysis to the wintertime (JF) sea ice loss in the Barents-Kara seas. The review of prior literature on Arctic amplification and sea ice loss is often confusing, including in the definition of key concepts such as Arctic amplification or albedo feedback. Based on the explanations given in the manuscript, I cannot understand the authors' methodology sufficiently to judge its value. The manuscript lacks a critical appreciation of the method, e.g. a discussion of how much of the time-series and trend is actually captured by the first 'mode' obtained in the analysis. My fundamental concern with the manuscript is that it uses correlations to establish causalities and feedbacks, with little regard to the physical and meteorological phenomena discussed. As an example, the feedback loop suggested as a key result of the paper begins with sea ice reduction which supposedly causes warming of 850 hPa temperatures. The alternative explanation that warm air advection contributes to sea ice loss is

at least as plausible, bot not even mentioned in the manuscript. I further do not see any justification for fitting an exponential to the time series of sea ice loss in the Barents-Kara seas, and far less for using that fit to make a prediction on when this ocean area would remain ice-free in winter.

In conclusion, I regret to say that the manuscript fails to meet basic scientific standards.

---

## Author Comment (AC1) · 7 Jul 2017

Response to the Comments of Reviewer 1

The manuscript attempts to address the mechanisms of Arctic amplification of climate warming and sea ice loss. Cyclostationary empirical orthogonal functions are applied on ERA-Interim reanalysis products, and the methodology includes novel aspects. Some interesting results are found on the relationships between turbulent surface fluxes, longwave radiation, and sea ice loss. After substantial revisions the manuscript has potential for a good paper in The Cryosphere.

Major comments

Comment1(C1): The authors focus on the statistical relationships of the spatial pat-

terns of anomalies in wintertime sea ice concentration, turbulent surface fluxes of sensible and latent heat, upward and downward longwave radiation, as well as air temperature, humidity and total cloud cover. Both the Arctic amplification and sea ice loss are, however, much more complicated processes, involving many factors, such as the large-scale atmospheric transports of heat and moisture from lower latitudes to the Arctic, and oceanic transports of heat and freshwater. Also, the role of clouds in the Arctic climate system cannot be characterized simply by the total cloud cover; the cloud water and ice contents are at least equally important, as are also the complex interactions between the surface fluxes, boundary-layer turbulence, cloud physics and radiative transfer. I don't mean that the authors should address all these processes, but they should make it clear in the manuscript that they restrict to processes acting in the Arctic, ignoring the forcing from lower latitudes, and they should pay more attention to cloud water and ice contents, which are physically more meaningful variables than the total cloud cover based on reanalysis.

Response1(R1): We thank the reviewer for detailed and constructive comments on the manuscript. As the reviewer mentioned, there are other processes, particularly forcing from lower latitudes, which are important for Arctic amplification and sea ice reduction. As can be seen in Figure R1, there is a net convergence of moisture transport and heat transport over the region of sea ice reduction, although the center of action is over the Greenland Sea. Thus, moisture and heat transport from lower latitudes apparently affects the variation of sea ice concentration. Figure R2 further shows that there is an appreciable correlation between the specific humidity variation and convergence of moisture transport (corr=0.62) and between the lower tropospheric temperature and convergence of heat transport (corr=0.33). Thus, it seems that both the convergence of moisture transport and the convergence of heat transport are at least partly responsible for the variation of specific humidity and temperature in the lower troposphere. On the other hand, the convergence of horizontal transport of moisture cannot explain one essential element of specific humidity anomaly - the mean of anomalous specific humidity. As can be seen in Figure R2a, the mean of moisture convergence is close to

$0.6 \times 10^{-6}$ g/kg/sec, which amounts to $\sim 0.05$ g/kg of moisture. This value explains only about 17% of the mean value of anomalous specific humidity ($\sim 0.3$ g/kg); the remainder should derive from a vertical process. Consider the following moisture conservation equation:

$$\frac{\partial q}{\partial t} = -\vec{u} \cdot \nabla q + S \doteq -\nabla \cdot (q\vec{u}) + S = -\nabla_h \cdot (q\vec{u}) - \frac{\partial (qw)}{\partial z} + S.$$

The convergence of the horizontal moisture transport is not so effective as the convergence of the vertical moisture transport in the equation above. As can be seen in Fig. R3, the anomalous evaporation due to sea ice reduction is positive throughout the winter and its magnitude is reasonable in comparison with the increase in specific humidity. The two time series in Fig. R3 are negatively correlated (except for the mean), indicating that increase (decrease) in specific humidity due to positive (negative) convergence of moisture transport decreases (increases) evaporation from the surface of the ocean; this is a reasonable explanation according to the bulk formula.

Likewise, the variation of the thermal advection and the subsequent convergence of the heat flux are highly correlated with the variation of downward longwave radiation and the lower tropospheric (850 hPa) temperature (see Fig. R2b). On the other hand, the small mean value of the convergence of the horizontal heat flux cannot explain the significant nonzero mean of the anomalous downward longwave radiation or the anomalous lower tropospheric (850 hPa) temperature. Thus, we think that the vertical process should be invoked to account for the significant changes in the means of the variables over the Barents-Kara Seas.

This is a serious issue and requires more detailed calculation and convincing demonstration, which we do not wish to pursue in the present study. We, however, acknowledge that we restrict ourselves to processes acting in the Arctic, ignoring the forcing from lower latitudes. [P3 L4-5: It should be noted that our discussion is restricted to processes in the Arctic; forcing from lower latitudes can also be important in the

process of Arctic amplification and sea ice reduction.]

We showed the pattern of total cloud cover, since several authors address that radiative forcing produced by clouds is an important mechanism for Arctic amplification. As the reviewer mentioned, cloud liquid water and cloud ice water may be a better measure of the effect of clouds rather than total cloud cover. Figure R4 indeed shows the patterns of total cloud liquid water and total cloud ice water exhibit better consistency with the region of sea ice reduction. In lieu of this new finding, we will replace the pattern of total cloud cover by the patterns of TCLW and TCIW in Fig. 4. [P5 L17-21: The patterns of total cloud liquid water and total cloud ice water, which are the key variables for the formation of clouds, also exhibit a strong response over the region of sea ice reduction although their centers of action are shifted toward the Greenland Sea (Fig. 4d). The pattern of total cloud cover, however, does not show any strong cloud activity over the region of sea ice reduction (Fig. S3 in the supplementary information); it should be understood that cloud cover is a difficult variable to simulate accurately in a reanalysis model.]

C2: Atmospheric reanalyses include serious errors in the Polar regions. This is the case particularly for surface fluxes and near-surface meteorological variables (e.g. Jakobson et al., 2012; Tastula et al., 2013). Hence, to obtain more robust results, I suggest to repeat the calculations using a second reanalysis (e.g. NCEP-CFSR) in addition to ERA-Interim.

R2: In response to the reviewer's suggestion, we analyzed a limited number of variables from the NCEP reanalysis product in order to reproduce the key results in the present manuscript. Figure R5 shows the regressed loading vectors derived from the 1979-2016 NCEP reanalysis product with the sea ice loss mode as the target variable. As a comparison between Figs. R5 and R6 shows, there is no essential difference between the two sets of regressed loading vectors except for a small difference in the scales. This magnitude difference seems to be due to slightly different sensitivity of sea ice to atmospheric and oceanic forcing in the two datasets. This exercise confirms

that the behavior of the atmospheric variables in association with the sea ice reduction in the Barents-Kara Seas is not significantly different between the two reanalysis products and the physical mechanism addressed in the present study is not overly sensitive to the choice of a model dataset.

Figure R7 further shows the daily variation of surface (2 m) air temperature, 850 hPa air temperature, upward longwave radiation and downward longwave radiation over the region of sea ice reduction (21°-79.5°E × 75°-79.5°N). As a comparison between Fig. R7 and Fig. 6 in the manuscript shows, the daily variation of derived from the NCEP reanalysis product is fairly similar to that derived from the ERA-Interim product. Again, there is a slight difference in the scales of anomalous variable, but the daily variation derived from the two datasets is not much different (see Figure 6 in the manuscript) confirming that the physical mechanism addressed in the present study is not sensitive to the choice of a dataset. We added the following statement. [P8 L8-11: Finally, it should be mentioned that the feedback process does not seem to be sensitive to a choice of the dataset. A similar experiment conducted by using the NCEP reanalysis data produces essentially identical results except for a slight overestimation of the strength of the anomalous patterns in Fig. 1a-f and Fig. 6 (see Figs. S5 and S6 in the supplementary information).]

C3: Several results detected from the reanalysis require a better physical explanation.

C3a: Lines 32-33: Why does the region of sea ice loss generate anticyclonic circulation? Figure 3 is not clear in this respect. Show maps of sea level pressure or geopotential height at a relevant pressure level to illustrate this and explain the physical mechanism resulting in an anticyconic circulation.

R3a: Figures R8a and R8b show the winter-averaged patterns of SAT, and lower-tropospheric circulation associated with the sea ice loss mode. Figures R8c and R8d show the lower tropospheric vertical sections of temperature, geopotential height, and wind along 60°E and 80°N across the center of action. It is difficult to explain what exactly is happening dynamically based on data analysis alone. Nonetheless, the physical variables in Fig. R8 are physically consistent with each other. For example, the lower tropospheric wind field seems nearly in geostrophic balance with the geopotential height field. Further, Fig. R9 shows that the anomalous geopotential height field is nearly in hydrostatic balance with the anomalous temperature field:

$$(dZ)_j = -(RT_j)/g(d\ln p)_j,$$

where

$$(dZ)_j = Z_j - Z_{(j-1)}, T_j = ((T_j + T_{(j-1)}))/2, (d\ln p)_j = \ln p_j - \ln p_{(j-1)}.$$

Thus, it seems that the release of energy in the form of radiation and heat flux changes the temperature, and geopotential height in the lower troposphere adjusts in accordance with the hydrostatic balance. [new Figure 3: We replaced the 925 hPa air temperature pattern by the lower tropospheric geopotential height and wind pattern.] [P5 L6-8: Figure 3 shows the anomalous surface (2 m) air temperature, the lower tropospheric geopotential height and wind and the vertical section of anomalous temperature, geopotential height and wind along 60°E and 80°N associated with sea ice reduction.] [P5 L10-11: ... consistent according to the hydrostatic equation (see Fig. S2).] [We added relevant discussion in conjunction with Fig. R9 in the supplementary information together with the figure (see Figure S2 and corresponding explanation).]

C3b: Page 5, lines 29-30: if the turbulent surface fluxes are upward and net longwave radiation is upwards, they tend to reduce the Earth surface temperature but increase the near-surface air temperature, not decrease it.

R3b: We used the surface (2 m) air temperature as a proxy for the surface temperature, since there is no surface temperature variable in the ERA-Interim reanalysis product. Thus, we assumed that the atmosphere up to 2 m height from the surface essentially

behaves the same way as the surface with a negligible absorption of turbulent heat flux. If the anomalous 2 m air temperature is not significantly different that of surface temperature, the amount of anomalous net longwave radiation at 2 m level would not be much different from that at the surface. Therefore, there would be net deficit of radiation energy at the 2 m level, resulting in a decrease in surface air temperature. As the reviewer indicated, however, surface turbulent fluxes may be consumed to raise the 2 m air temperature although we do not know the amount of energy consumed at this level. Thus, we changed the sentence as follows: [P6 L8-9: This implies that surface air temperature should decrease, preventing further sea ice reduction.] See our discussion in Part (c) for more details.

C3c1: Page 6, lines 6-7: Why does a change in 2 m air temperature slightly lead the upward longwave radiation? Further, on lines 8-9: an increase in 2-m air temperature does not have a causal effect of increasing the upward longwave radiation (a statistical relationship may naturally exist). The upward longwave radiation at the surface (which is the product archived in ERA-Interim) is controlled by the surface temperature (and emissivity), not by the 2-m air temperature.

R3c1: It is difficult to answer why there is a lag between the two variables. In each grid box, upward longwave radiation is computed via (see Fig. R10)

$$ULR(t) = \epsilon_i(t)R(T_i(t))f_i(t) + \epsilon_0(t)R(T_0(t))f_0(t), \qquad (1)$$

where $R(t)$ is radiation as a function of radiating temperature $T$, $\epsilon$ is emissivity, $f$ is a fractional area, the subscripts $i$ and $o$ stand for the "ice-covered" and "open (ice free)" areas, respectively, and $ULR(t)$ denotes the averaged upward longwave radiation in the grid box. In (1), the radiating function (basically Planck function) is a nonlinear function of temperature, and the emissivity $\epsilon$ may be dependent upon the sky condition as well as the surface condition. Further, $f$ varies in time. Thus, the calculation of upward longwave radiation in each grid box may not be a linear function of (grid-averaged)

surface temperature. This means that anomalous upward longwave radiation is not a linear function of anomalous surface temperature. Also note that the amount of anomalous radiation is not a function of anomalous temperature; it is determined by the temporally varying background (mean) temperature plus the anomalous temperature. Therefore, the amount of anomalous upward longwave radiation is not simply a function of anomalous surface temperature.

The 2 m air temperature change slightly leads the change in upward longwave radiation according to our lagged correlation analysis based on the 3-hourly ERA-Interim data. As can be seen in our original Figure 7 in the manuscript, however, the lead is less than 0.1 days (less than one time step) and the lagged correlation varies little between the lag range of [0, 0.1] day. Thus, we cannot confirm if this lead/lag relationship is a realistic relationship between the two variables or an artifact of analysis. Based on the shape of the lagged correlation (correlation for positive lags is generally stronger than that for negative lags at the same distance from lag 0), we thought that it was reasonable to say that 2m air temperature slightly leads upward longwave radiation.

We used the 2 m air temperature as a proxy for the surface temperature in order to explain changes in upward longwave radiation. As the reviewer mentioned, surface temperature instead of surface air temperature should be used in order to address this issue. Unfortunately, there is no variable called the "surface temperature" in the ERA-Interim product. We could have used the skin temperature $T_{SK}$ but it is defined as the temperature of the surface at radiative equilibrium:

$$R_{SW} + R_{LW} + J_s + LJ_q = \Lambda_{skin}(T_{SK} - T_S), \qquad (2)$$

where $R_{SW}$ and $R_{LW}$ are shortwave and longwave radiation fluxes at the surface, $J_s$ and $J_q$ are heat and moisture fluxes, $\Lambda_{skin}$ is skin conductivity, $T_S$ is temperature of soil, snow or ice. It is a poor reflection of radiative energy surplus or deficit at the surface, since any surplus/deficit of radiative energy is compensated by turbulent fluxes and/or

heat conduction between the level of skin temperature and the underlying soil, snow or ice. Further, skin temperature is identical with sea surface temperature over ice-free regions. Thus, change in net longwave radiation is not reflected in the skin temperature over the open ocean.

The use of skin temperature does not improve the interpretation of the analysis results. As seen in Table R1, skin temperature is slightly better correlated with upward longwave radiation, but exhibits much poorer correlation with downward longwave radiation, net longwave radiation or 850 hPa air temperature for the sea ice loss mode. As seen in Fig. R11, the spatial patterns of the two variables are essentially identical particularly over the region of sea ice reduction in the Barents-Kara Seas. Thus, the use of skin temperature instead of surface air temperature does not alter or improve the physical interpretation of the analysis results in the present study. Neither could we confirm that surface air temperature leads the upward longwave radiation, nor could we find a suitable variable to replace the surface temperature. Therefore, only option we have is to remove the statement about the lead/lag relationship between the surface air temperature and the upward longwave radiation. We modified Figs. 7b and 8, and the corresponding discussion as follows. [P6 L21-22: ... downward longwave radiation, which leads to a sea ice reduction. As a result, surface temperature and upward longwave radiation may increase.] [new Figure 7b  Figure 8]

Table R1. Correlation of the loading vectors of 850 hPa air temperature (T850), upward longwave radiation (ULW), downward longwave radiation (DLW), and net longwave radiation (NLW) with surface air temperature (SAT) in the second column and with skin temperature (SKT) in the third column.

|     | T850  | ULW   | DLW   | NLW   |
|-----|-------|-------|-------|-------|
| SAT | 0.861 | 0.878 | 0.916 | 0.681 |
| SKT | 0.448 | 0.891 | 0.722 | 0.286 |

[Figure]

C3c2: Further, on line 10: Instead of a causal effect from upward longwave radiation to sea ice concentration change, a reduction of sea ice concentration in winter must have an immediate effect in strongly increasing the upward longwave radiation.

R3c2: We agree with the reviewer. We used the increased upward longwave radiation as an evidence of the increased surface temperature, which eventually leads to sea ice reduction. As the reviewer mentioned, however, sea ice reduction immediately increases the upward longwave radiation because of the exposure of higher sea surface temperature. Thus, the lead/lag relationship between the upward longwave radiation and sea ice reduction is not so straightforward. We removed the following sentence: [P6 L22: Further, both downward and upward longwave radiation changes seem to lead sea ice concentration change.] At the same time, we modified Figure 8 to reflect this change. [new Figure 8]

C3d: Page 6, line 13: related to the above, I suggest removing the words "surface air temperature increases and".

R3d: We followed the suggestion of the reviewer. [P6 L25: As a result, surface air temperature increases and sea ice melts]

Minor comments

C1: Page 2, line 7: I think Vihma (2014) should be dropped from this line.

R1: We removed the Vihma (2014) reference. [P2 L7: Vihma (2014) is removed now.]

C2: Page 2, lines 13-17: During spring and early summer the albedo decreases from roughly 0.85 of dry snow-covered ice to 0.4 of melting ice. Hence, the albedo feedback is important already during the snow and ice melt, already before the appearance of open sea.

R2: We modified the sentence as follows: [P2 L17-18: . . . absorbing atmospheric heat during summer. The albedo feedback is also important during the snow and ice melt in spring and early summer even before the appearance of open sea.]

C3: Page 2, line 18: the term "oceanic heat transport" is not the best possible, as it may be interpreted as the horizontal transport from lower latitudes to the Arctic.

R3: We modified the sentence as follows: [P2 L19-20: ... mechanism becomes oceanic horizontal advection and vertical convection of heat (Screen and Simmonds, 2010b).]

C4: Equation 1 and the text below: explain what is r.

R4: We modified the sentence as follow: [P3 L16: ... physical process, describes ... on a longer time scale, and $r$ and $t$ denote location and time, respectively.]

C5: Page 4, line 23: On the basis of Figure 2, I would not write that the sea ice concentration remains nearly stationary throughout the winter.

R5: We modified the sentence as follows: [P5 L1-2: Sea ice concentration varies slightly on a daily basis, and its fluctuation is less than 2% from the mean value of −14.7% throughout the winter (Fig. 2).]

C6: Page 4, line 25: I cannot detect the 40% decrease from the amplitude time series. Also, better explain what the amplitude represents.

R6: As shown in (1), actual data is obtained by multiplying each loading vector with corresponding PC (amplitude) time series. According to Fig. 1g, the amplitude time series has increased by about 2.6 during the 37-year period. Multiplying this value with 14.7% (loading vector; Fig. 2), we obtain $\sim 38.2\%$. Actually, Fig. 1h is obtained by multiplying the PC time series Fig. 1g with the corresponding loading vector Fig. 2. We modified the sentence as follows: [P5 L3-5: Multiplying the amplitude (PC) time series (Fig. 1g) with the loading vector (Fig. 2) of the sea ice loss mode as in (1), actual sea ice concentration time series is obtained as in Fig. 1h. According to Fig. 1h, sea ice concentration has decreased by $\sim 40\%$ during the last 37 years.] [Figure 1 caption: ... The red curve in Fig. 1h is obtained by multiplying the loading vector of sea ice concentration (Fig. 1a) averaged in the boxed area with the amplitude time series (Fig.

1g) according to (1). . . .]

C7: Page 5, line 9: "the little connectivity" between sea ice reduction and total cloud cover may originate from the fact that sea ice reduction generates two effects that compete against each other: increased latent heat flux tends to increase cloudiness but increasing sensible heat flux tends to reduce it.

R7: We thank the reviewer for enlightening us. As mentioned in our response to your major comments, we included the total cloud pattern since several authors mention the possible role of clouds for Arctic amplification. See our response to Major Comment 1 above.

C8: Page 6, lines 23-25: These are interesting numbers. Please, confirm if these are winter means in 1979-2016 averaged over sea areas north of 60N. It might be interesting to compare them against results of Lupkes et al. (2008), which show air temperature responses to 1% reduction in sea ice cover in different conditions.

R8: It is the result based on the average over the region of sea ice reduction ($21°$-$79.5°$E $\times$ $75°$-$79.5°$N) in the Barents-Kara Seas; it shows the values of anomalous radiation and surface fluxes for an average sea ice reduction of $\sim 15\%$ (see Fig. 2). Lüpkes et al. (2008) conducted experiments in different settings using a 1D atmospheric model coupled with snow/sea ice model. Therefore, a rigorous comparison is impossible. Our numbers are smaller than those in Lüpkes et al. but are of the same order of magnitude. It is difficult to explain the reasons for this difference, but the absence of horizontal advection is a plausible cause. In the presence of horizontal advection, anomalous temperature and fluxes over sea ice leads are quickly diffused, resulting in reduced local maxima. We can see a hint of horizontal advection in Fig. 1 in the manuscript; while turbulent heat flux is nearly confined to the area of sea ice loss, 2 m air temperature and other variables are smoothed out over a much wider area. We would rather not include this discussion in the revised text, since we eliminated 2 m air temperature increase in the feedback loop. Moreover, our explanation above is

somewhat premature and conjectural in nature. [no modification]

C9: Figure 2. Is plot (a) needed at all, as the same line appears in plot (b)?

R9: We removed Figure 2a. [new Figure 2]

C10: Figure 4. Add information on colour scales and absolute values. Only the contour intervals are given.

R10: We revised Figure 4 in order to provide the necessary information the reviewer asked. [new Figure 4; figure caption: The red contour is drawn at the value of the contour interval.]

C11: Figure 5. Does the shading represent sea ice concentration in all four plots? If yes, why it includes small differences between the plots?

R11: We accidentally used different shading interval for the sea ice concentration field in Fig. 5c. We corrected the figure. [new Figure 5]

C12: Figure 6. Explain better how the time series in days should be interpreted. It cannot be the mean over 1979-2016. Is it from some selected year?

R12: It is the plot of regressed loading vector $B_1(r, t)$ of the sea ice loss mode averaged over the region of sea ice loss (21°-79.5°E × 75°-79.5°N) for different variables. The daily time series are interpreted as typical winter variation of surface fluxes and radiation associated with the sea ice reduction in Fig. 2. Actual data associated with the sea ice loss mode is obtained by multiplying the loading vector with the corresponding PC time series, i.e., the space-time evolution pattern associated with the sea ice loss mode is $T^{(1)}(r, t) = B_1(r, t)T_1(t)$. Thus, the typical time series of surface fluxes and radiation depicted in Fig. 6 are amplifying according to Fig. 1g. We added more description on this figure. [P5 L28-30: Figure 6 shows the winter daily variations of the regressed loading vectors in (6) (terms in curly braces) averaged over the region of sea ice reduction (21°-79.5° E × 75°-79.5° N); it may be interpreted as the atmospheric response to the sea ice reduction shown in Fig. 2.]

C13: Figure 8. Referring to my previous comments, I suggest dropping "2m T increase (0.24)" from the figure, and drawing an arrow directly from increased LW-down to sea ice reduction. Otherwise, provide a good explanation on the causality of the link.

R13: We followed the suggestion of the reviewer. [new Figure 8]

** The combined response file including a marked-up manuscript is attached.

References: Jakobson, E., T. Vihma, T. Palo, L. Jakobson, H. Keernik, and J. Jaagus (2012). Validation of atmospheric reanalyzes over the central Arctic Ocean, Geophys. Res. Lett. 39, L10802, doi:10.1029/2012GL051591. Tastula, E.-M., T. Vihma, E. L. Andreas, and B. Galperin (2013), Validation of the diurnal cycles in atmospheric reanalyses over Antarctic sea ice, J. Geophys. Res. Atmos., 118, 4194–4204, doi:10.1002/jgrd.50336.

Please also note the supplement to this comment:
https://www.the-cryosphere-discuss.net/tc-2017-39/tc-2017-39-AC1-supplement.pdf

[Figure]

[Figure]

**Fig. 1.** Winter-averaged (left panel) moisture transport (streamline) and its convergence (shade) and (right panel) heat transport (streamline) and its convergence (shade) in the lower troposphere

[Figure]

**Fig. 2.** The daily time series of anomalous specific humidity and anomalous moisture convergence averaged over the sea ice loss region (21°-79.5°E × 75°-79.5°N) in the Barents-Kara Seas. These ti

**Fig. 3.** The daily variation of specific humidity (red) and evaporation (blue) averaged over the region of sea ice reduction (21°-79.5°E × 75°-79.5°N) in the Barents-Kara Seas.

TCLW          TCIW

−10 −5 −3 −1.5 −0.5 0 0.5 1.5 3 5 10

**Fig. 4.** The winter-averaged patterns of total cloud liquid water (left) and total cloud ice water (right) associated with the sea ice loss mode.

[Figure]

**Fig. 5.** The regressed patterns of atmospheric variables based on the NCEP Reanalysis product (1979-2016). The target is the sea ice loss mode.

[Figure]

**Fig. 6.** The regressed patterns of atmospheric variables based on the ERA-Interim reanalysis product (Figure 1a-f in the manuscript).

[Figure]

**Fig. 7.** Daily pattern of variability over the region of sea ice loss (21°-79.5°E × 75°-79.5°N) derived from the NCEP reanalysis data: (a) 2 m air temperature (red), 850 hPa air temperature × 2 (black), and u

(a) 2m AIR T  (0.5° C)

(b) Z & (U,V)

(c) 60° E

(d) 80° N

**Fig. 8.** The spatial patterns (60° –90°N) of (a) 2 m air temperature, and (b) lower tropospheric (1000-900 hPa) geopotential height (red contour) and wind (streamline). (c and d) The lower tropospheric vertic

[Figure]

**Fig. 9.** The pressure layer thickness ($\Delta Z = Z(p\_1) - Z(p\_0)$) derived from the geopotential height pattern in Fig. R8 (shade) and that derived from the hydrostatic equation (contour). The red contour represents

[Figure]

radiation

emissivity &
temperature

**Fig. 10.** A typical situation of calculating upward longwave radiation in each grid box. Upward longwave radiation is calculated for sea ice tile and open ocean tile separately in order to calculate total upwa

[Figure]

**Fig. 11.** The regressed patterns of surface air temperature (SAT) and skin temperature (SKT) with the sea ice loss mode as the target (contour).

**Supplement:**

**Response to the Comments of Reviewer #1**

The manuscript attempts to address the mechanisms of Arctic amplification of climate warming and sea ice loss. Cyclostationary empirical orthogonal functions are applied on ERA-Interim reanalysis products, and the methodology includes novel aspects. Some interesting results are found on the relationships between turbulent surface fluxes, longwave radiation, and sea ice loss. After substantial revisions the manuscript has potential for a good paper in The Cryosphere.

Major comments

Comment1(C1): The authors focus on the statistical relationships of the spatial patterns of anomalies in wintertime sea ice concentration, turbulent surface fluxes of sensible and latent heat, upward and downward longwave radiation, as well as air temperature, humidity and total cloud cover. Both the Arctic amplification and sea ice loss are, however, much more complicated processes, involving many factors, such as the large-scale atmospheric transports of heat and moisture from lower latitudes to the Arctic, and oceanic transports of heat and freshwater. Also, the role of clouds in the Arctic climate system cannot be characterized simply by the total cloud cover; the cloud water and ice contents are at least equally important, as are also the complex interactions between the surface fluxes, boundary-layer turbulence, cloud physics and radiative transfer. I don't mean that the authors should address all these processes, but they should make it clear in the manuscript that they restrict to processes acting in the Arctic, ignoring the forcing from lower latitudes, and they should pay more attention to cloud water and ice contents, which are physically more meaningful variables than the total cloud cover based on reanalysis.

Response1(R1): We thank the reviewer for detailed and constructive comments on the manuscript. As the reviewer mentioned, there are other processes, particularly forcing from lower latitudes, which are important for Arctic amplification and sea ice reduction. As can be seen in Figure R1, there is a net convergence of moisture transport and heat transport over the region of sea ice reduction, although the center of action is over the Greenland Sea. Thus, moisture and heat transport from lower latitudes apparently affects the variation of sea ice concentration. Figure R2 further shows that there is an appreciable correlation between the specific humidity variation and convergence of moisture transport (corr=0.62) and between the lower tropospheric temperature and convergence of heat transport (corr=0.33). Thus, it seems that both the convergence of moisture transport and the convergence of heat transport are at least partly responsible for the variation of specific humidity and temperature in the lower troposphere. On the other hand, the convergence of horizontal transport of moisture cannot explain one essential element of specific humidity anomaly—the mean of anomalous specific humidity. As can be seen in Figure R2a, the mean of moisture convergence is close to $0.6\times10^{-6}$ g/kg/sec, which amounts to ~0.05 g/kg of moisture. This value explains only about 17% of the mean value of anomalous specific humidity (~0.3 g/kg); the remainder

[Figure]

Figure R1. Winter-averaged (left panel) moisture transport (streamline) and its convergence (shade) and (right panel) heat transport (streamline) and its convergence (shade) in the lower troposphere (1000-850 hPa) associated with the sea ice loss mode.

should derive from a vertical process. Consider the following moisture conservation equation:

$$\frac{\partial q}{\partial t} = -\vec{u} \cdot \nabla q + S \doteq -\nabla \cdot (q\vec{u}) + S = -\nabla_h \cdot (q\vec{u}) - \frac{\partial(qw)}{\partial z} + S.$$

[Figure]

Figure R2. The daily time series of anomalous specific humidity and anomalous moisture convergence averaged over the sea ice loss region (21°-79.5°E × 75°-79.5°N) in the Barents-Kara Seas. These time series are derived from the regressed loading vectors associated with the sea ice loss mode.

[Figure]

Figure R3. The daily variation of specific humidity (red) and evaporation (blue) averaged over the region of sea ice reduction (21°-79.5°E × 75°-79.5°N) in the Barents-Kara Seas.

The convergence of the horizontal moisture transport is not so effective as the convergence of the vertical moisture transport in the equation above. As can be seen in Fig. R3, the anomalous evaporation due to sea ice reduction is positive throughout the winter and its magnitude is reasonable in comparison with the increase in specific humidity. The two time series in Fig. R3 are negatively correlated (except for the mean), indicating that increase (decrease) in specific humidity due to positive

10 (negative) convergence of moisture transport decreases (increases) evaporation from the surface of the ocean; this is a reasonable explanation according to the bulk formula.

Likewise, the variation of the thermal advection and the subsequent convergence of the heat flux are highly correlated with the variation of downward longwave radiation and the lower tropospheric (850 hPa) temperature (see Fig. R2b). On the

15 other hand, the small mean value of the convergence of the horizontal heat flux cannot explain the significant nonzero mean of the anomalous downward longwave radiation or the anomalous lower tropospheric (850 hPa) temperature. Thus, we think that the vertical process should be invoked to account for the significant changes in the means of the variables over the Barents-Kara Seas.

20 This is a serious issue and requires more detailed calculation and convincing demonstration, which we do not wish to pursue in the present study. We, however, acknowledge that we restrict ourselves to processes acting in the Arctic, ignoring the forcing from lower latitudes. [P3 L4-5: It should be noted that our discussion is restricted to processes in the Arctic; forcing from lower latitudes can also be important in the process of Arctic amplification and sea ice reduction.]

25 We showed the pattern of total cloud cover, since several authors address that radiative forcing produced by clouds is an important mechanism for Arctic amplification. As the reviewer mentioned, cloud liquid water and cloud ice water may be a better measure of the effect of clouds rather than total cloud cover. Figure R4 indeed shows the patterns of total cloud liquid

[Figure]

Figure R4. The winter-averaged patterns of total cloud liquid water (left) and total cloud ice water (right) associated with the sea ice loss mode.

water and total cloud ice water exhibit better consistency with the region of sea ice reduction. In lieu of this new finding, we will replace the pattern of total cloud cover by the patterns of TCLW and TCIW in Fig. 4. [P5 L17-21: The patterns of total cloud liquid water and total cloud ice water, which are the key variables for the formation of clouds, also exhibit a strong response over the region of sea ice reduction although their centers of action are shifted toward the Greenland Sea (Fig. 4d). The pattern of total cloud cover, however, does not show any strong cloud activity over the region of sea ice reduction (Fig. S3 in the supplementary information); it should be understood that cloud cover is a difficult variable to simulate accurately in a reanalysis model.]

C2: Atmospheric reanalyses include serious errors in the Polar regions. This is the case particularly for surface fluxes and near-surface meteorological variables (e.g. Jakobson et al., 2012; Tastula et al., 2013). Hence, to obtain more robust results, I suggest to repeat the calculations using a second reanalysis (e.g. NCEP-CFSR) in addition to ERA-Interim.

R2: In response to the reviewer's suggestion, we analyzed a limited number of variables from the NCEP reanalysis product in order to reproduce the key results in the present manuscript. Figure R5 shows the regressed loading vectors derived from the 1979-2016 NCEP reanalysis product with the sea ice loss mode as the target variable. As a comparison between Figs. R5 and R6 shows, there is no essential difference between the two sets of regressed loading vectors except for a small difference

[Figure]

Figure R5.  The regressed patterns of atmospheric variables based on the NCEP Reanalysis product (1979-2016).  The target is the sea ice loss mode.

[Figure]

(a) SIC (2 %) & 2m AIR T (0.5° C)

(b) 1000-850 hPa SH (3×10⁻² g Kg⁻¹)

(c) ULW at SFC (2 W m⁻²)

(d) DLW at SFC (2 W m⁻²)

(e) TURBULENT FLUX (4 W m⁻²)

(f) 850 hPa T (0.2° C)

Figure R6. The regressed patterns of atmospheric variables based on the ERA-Interim reanalysis product (Figure 1a-f in the manuscript).

[Figure]

Figure R7. Daily pattern of variability over the region of sea ice loss (21°-79.5°E × 75°-79.5°N) derived from the NCEP reanalysis data: (a) 2 m air temperature (red), 850 hPa air temperature × 2 (black), and upward longwave radiation (blue), and (b) same as (a) except for the regressed downward longwave radiation (blue). The straight line represents the winter mean value of anomalous 2 m air temperature. Correlation of upward and downward longwave radiation with 2 m air temperature is respectively 0.95 and 0.94, whereas correlation with 850 hPa air temperature is respectively 0.81 and 0.86.

in the scales. This magnitude difference seems to be due to slightly different sensitivity of sea ice to atmospheric and oceanic forcing in the two datasets. This exercise confirms that the behavior of the atmospheric variables in association with the sea ice reduction in the Barents-Kara Seas is not significantly different between the two reanalysis products and the physical mechanism addressed in the present study is not overly sensitive to the choice of a model dataset.

Figure R7 further shows the daily variation of surface (2 m) air temperature, 850 hPa air temperature, upward longwave radiation and downward longwave radiation over the region of sea ice reduction (21°-79.5°E × 75°-79.5°N). As a comparison between Fig. R7 and Fig. 6 in the manuscript shows, the daily variation of derived from the NCEP reanalysis product is fairly similar to that derived from the ERA-Interim product. Again, there is a slight difference in the scales of anomalous variable, but the daily variation derived from the two datasets is not much different (see Figure 6 in the manuscript) confirming that the physical mechanism addressed in the present study is not sensitive to the choice of a dataset. We added the following statement. [P8 L8-11: Finally, it should be mentioned that the feedback process does not seem to be sensitive to a choice of the dataset. A similar experiment conducted by using the NCEP reanalysis data produces essentially identical results except for a slight overestimation of the strength of the anomalous patterns in Fig. 1a-f and Fig. 6 (see Figs. S5 and S6 in the supplementary information).]

C3: Several results detected from the reanalysis require a better physical explanation.

C3a: Lines 32-33: Why does the region of sea ice loss generate anticyclonic circulation? Figure 3 is not clear in this respect. Show maps of sea level pressure or geopotential height at a relevant pressure level to illustrate this and explain the physical mechanism resulting in an anticyconic circulation.

R3a: Figures R8a and R8b show the winter-averaged patterns of SAT, and lower-tropospheric circulation associated with the sea ice loss mode. Figures R8c and R8d show the lower tropospheric vertical sections of temperature, geopotential height,

[Figure]

Figure R8. The spatial patterns (60°–90°N) of (a) 2 m air temperature, and (b) lower tropospheric (1000-900 hPa) geopotential height (red contour) and wind (streamline). (c and d) The lower tropospheric vertical sections of temperature (shade), geopotential height (black contour) and wind (blue contour) along 60°E and 80°N.

[Figure]

Figure R9. The pressure layer thickness ($\Delta Z = Z(p_1) - Z(p_0)$) derived from the geopotential height pattern in Fig. R8 (shade) and that derived from the hydrostatic equation (contour). The red contour represents the thickness of 1.5 m. The level $p_1$ is the level used for plotting and $p_0$ is the pressure level below $p_1$ at the interval of 25 hPa.

and wind along 60°E and 80°N across the center of action. It is difficult to explain what exactly is happening dynamically based on data analysis alone. Nonetheless, the physical variables in Fig. R8 are physically consistent with each other. For example, the lower tropospheric wind field seems nearly in geostrophic balance with the geopotential height field. Further, Fig. R9 shows that the anomalous geopotential height field is nearly in hydrostatic balance with the anomalous temperature field:

$$(dZ)_j = -\frac{R\langle T\rangle_j}{g}(d \ln p)_j,$$

where

$$(dZ)_j = Z_j - Z_{j-1}, \quad \langle T\rangle_j = \left(T_j + T_{j-1}\right)/2, \quad (d \ln p)_j = \ln p_j - \ln p_{j-1}.$$

Thus, it seems that the release of energy in the form of radiation and heat flux changes the temperature, and geopotential height in the lower troposphere adjusts in accordance with the hydrostatic balance. [new Figure 3: We replaced the 925 hPa air temperature pattern by the lower tropospheric geopotential height and wind pattern.] [P5 L6-8: Figure 3 shows the anomalous surface (2 m) air temperature, the lower tropospheric geopotential height and wind and the vertical section of anomalous temperature, geopotential height and wind along 60°E and 80°N associated with sea ice reduction.] [P5 L10-11: … consistent according to the hydrostatic equation (see Fig. S2).] [We added relevant discussion in conjunction with Fig. R9 in the supplementary information together with the figure (see Figure S2 and corresponding explanation).]

C3b: Page 5, lines 29-30: if the turbulent surface fluxes are upward and net longwave radiation is upwards, they tend to reduce the Earth surface temperature but increase the near-surface air temperature, not decrease it.

R3b: We used the surface (2 m) air temperature as a proxy for the surface temperature, since there is no surface temperature
5  variable in the ERA-Interim reanalysis product. Thus, we assumed that the atmosphere up to 2 m height from the surface essentially behaves the same way as the surface with a negligible absorption of turbulent heat flux. If the anomalous 2 m air temperature is not significantly different that of surface temperature, the amount of anomalous net longwave radiation at 2 m level would not be much different from that at the surface. Therefore, there would be net deficit of radiation energy at the 2 m level, resulting in a decrease in surface air temperature. As the reviewer indicated, however, surface turbulent fluxes may
10  be consumed to raise the 2 m air temperature although we do not know the amount of energy consumed at this level. Thus, we changed the sentence as follows: [P6 L8-9: This implies that surface  temperature should decrease, preventing further sea ice reduction.] See our discussion in Part (c) for more details.

C3c: Page 6, lines 6-7: Why does a change in 2 m air temperature slightly lead the upward longwave radiation? Further, on
15  lines 8-9: an increase in 2-m air temperature does not have a causal effect of increasing the upward longwave radiation (a statistical relationship may naturally exist). The upward longwave radiation at the surface (which is the product archived in ERA-Interim) is controlled by the surface temperature (and emissivity), not by the 2-m air temperature.

R3c: It is difficult to answer why there is a lag between the two variables. In each grid box, upward longwave radiation is
20  computed via (see Fig. R10)

$$ULR(t) = \varepsilon_i(t)R\big(T_i(t)\big)f_i(t) + \varepsilon_0(t)R\big(T_0(t)\big)f_0(t), \tag{1}$$

[Figure]

25  Figure R10.  A typical situation of calculating upward longwave radiation in each grid box.  Upward longwave radiation is calculated for sea ice tile and open ocean tile separately in order to calculate total upward longwave radiation for the grid box.  The emissivity and fractional area of sea ice are functions of time.

where $R(t)$ is radiation as a function of radiating temperature $T$, $\varepsilon$ is emissivity, $f$ is a fractional area, the subscripts $i$ and $o$ stand for the "ice-covered" and "open (ice free)" areas, respectively, and $ULR(t)$ denotes the averaged upward longwave radiation in the grid box. In (1), the radiating function (basically Planck function) is a nonlinear function of temperature, and the emissivity $\varepsilon$ may be dependent upon the sky condition as well as the surface condition. Further, $f$ varies in time. Thus,

5   the calculation of upward longwave radiation in each grid box may not be a linear function of (grid-averaged) surface temperature. This means that anomalous upward longwave radiation is not a linear function of anomalous surface temperature. Also note that the amount of anomalous radiation is not a function of anomalous temperature; it is determined by the temporally varying background (mean) temperature plus the anomalous temperature. Therefore, the amount of anomalous upward longwave radiation is not simply a function of anomalous surface temperature.

The 2 m air temperature change slightly leads the change in upward longwave radiation according to our lagged correlation analysis based on the 3-hourly ERA-Interim data. As can be seen in our original Figure 7 in the manuscript, however, the lead is less than 0.1 days (less than one time step) and the lagged correlation varies little between the lag range of [0, 0.1] day. Thus, we cannot confirm if this lead/lag relationship is a realistic relationship between the two variables or an artifact

15   of analysis. Based on the shape of the lagged correlation (correlation for positive lags is generally stronger than that for negative lags at the same distance from lag 0), we thought that it was reasonable to say that 2m air temperature slightly leads upward longwave radiation.

We used the 2 m air temperature as a proxy for the surface temperature in order to explain changes in upward longwave

20   radiation. As the reviewer mentioned, surface temperature instead of surface air temperature should be used in order to address this issue. Unfortunately, there is no variable called the "surface temperature" in the ERA-Interim product. We could have used the skin temperature $T_{SK}$ but it is defined as the temperature of the surface at radiative equilibrium:

$$R_{SW} + R_{LW} + J_s + LJ_q = \Lambda_{skin}(T_{SK} - T_S), \tag{2}$$

where $R_{SW}$ and $R_{LW}$ are shortwave and longwave radiation fluxes at the surface, $J_s$ and $J_q$ are heat and moisture fluxes, $\Lambda_{skin}$

25   is skin conductivity, $T_S$ is temperature of soil, snow or ice. It is a poor reflection of radiative energy surplus or deficit at the surface, since any surplus/deficit of radiative energy is compensated by turbulent fluxes and/or heat conduction between the level of skin temperature and the underlying soil, snow or ice. Further, skin temperature is identical with sea surface temperature over ice-free regions. Thus, change in net longwave radiation is not reflected in the skin temperature over the open ocean.

Table R1. Correlation of the loading vectors of 850 hPa air temperature (T_850), upward longwave radiation (ULW), downward longwave radiation (DLW), and net longwave radiation (NLW) with surface air temperature (SAT) in the second column and with skin temperature (SKT) in the third column.

|  | T_850 | ULW | DLW | NLW |
|---|---|---|---|---|
| SAT | 0.861 | 0.878 | 0.916 | 0.681 |
| SKT | 0.448 | 0.891 | 0.722 | 0.286 |

The use of skin temperature does not improve the interpretation of the analysis results. As seen in Table R1, skin temperature is slightly better correlated with upward longwave radiation, but exhibits much poorer correlation with downward longwave radiation, net longwave radiation or 850 hPa air temperature for the sea ice loss mode. As seen in Fig. R11, the spatial patterns of the two variables are essentially identical particularly over the region of sea ice reduction in the
10   Barents-Kara Seas. Thus, the use of skin temperature instead of surface air temperature does not alter or improve the physical interpretation of the analysis results in the present study. Neither could we confirm that surface air temperature leads the upward longwave radiation, nor could we find a suitable variable to replace the surface temperature. Therefore, only option we have is to remove the statement about the lead/lag relationship between the surface air temperature and the upward longwave radiation. We modified Figs. 7b and 8, and the corresponding discussion as follows. [P6 L21-22: …
15   downward longwave radiation, which leads to a sea ice reduction. As a result, surface temperature and upward longwave radiation may increase.] [new Figure 7b & Figure 8]

[Figure]

20   Figure R11. The regressed patterns of surface air temperature (SAT) and skin temperature (SKT) with the sea ice loss mode as the target (contour).

Further, on line 10: Instead of a causal effect from upward longwave radiation to sea ice concentration change, a reduction of sea ice concentration in winter must have an immediate effect in strongly increasing the upward longwave radiation.

We agree with the reviewer. We used the increased upward longwave radiation as an evidence of the increased surface

5 temperature, which eventually leads to sea ice reduction. As the reviewer mentioned, however, sea ice reduction immediately increases the upward longwave radiation because of the exposure of higher sea surface temperature. Thus, the lead/lag relationship between the upward longwave radiation and sea ice reduction is not so straightforward. We removed the following sentence: [P6 L22: ] At the same time, we modified Figure 8 to reflect this change. [new Figure 8]

C3d: Page 6, line 13: related to the above, I suggest removing the words "surface air temperature increases and".

R3d: We followed the suggestion of the reviewer. [P6 L25: As a result,  sea ice melts]

Minor comments

C1: Page 2, line 7: I think Vihma (2014) should be dropped from this line.

R1: We removed the Vihma (2014) reference. [P2 L7: Vihma (2014) is removed now.]

C2: Page 2, lines 13-17: During spring and early summer the albedo decreases from roughly 0.85 of dry snow-covered ice to 0.4 of melting ice. Hence, the albedo feedback is important already during the snow and ice melt, already before the
10 appearance of open sea.

R2: We modified the sentence as follows: [P2 L17-18: … absorbing atmospheric heat during summer. The albedo feedback is also important during the snow and ice melt in spring and early summer even before the appearance of open sea.]

15 C3: Page 2, line 18: the term "oceanic heat transport" is not the best possible, as it may be interpreted as the horizontal transport from lower latitudes to the Arctic.

R3: We modified the sentence as follows: [P2 L19-20: … mechanism becomes oceanic horizontal advection and vertical convection of heat (Screen and Simmonds, 2010b).]

C4: Equation 1 and the text below: explain what is r.

R4: We modified the sentence as follow: [P3 L16: … physical process, $T_n(t)$ describes … on a longer time scale, and $r$ and $t$ denote location and time, respectively.]

C5: Page 4, line 23: On the basis of Figure 2, I would not write that the sea ice concentration remains nearly stationary throughout the winter.

R5: We modified the sentence as follows: [P5 L1-2: Sea ice concentration varies slightly on a daily basis, and its
30 fluctuation is less than 2% from the mean value of −14.7% throughout the winter (Fig. 2).]

C6: Page 4, line 25: I cannot detect the 40% decrease from the amplitude time series. Also, better explain what the amplitude represents.

R6: As shown in (1), actual data is obtained by multiplying each loading vector with corresponding PC (amplitude) time series. According to Fig. 1g, the amplitude time series has increased by about 2.6 during the 37-year period. Multiplying this value with 14.7% (loading vector; Fig. 2), we obtain ~38.2%. Actually, Fig. 1h is obtained by multiplying the PC time series Fig. 1g with the corresponding loading vector Fig. 2. We modified the sentence as follows: [P5 L3-5: Multiplying the amplitude (PC) time series (Fig. 1g) with the loading vector (Fig. 2) of the sea ice loss mode as in (1), actual sea ice concentration time series is obtained as in Fig. 1h. According to Fig. 1h, sea ice concentration has decreased by ~40% during the last 37 years.] [Figure 1 caption: … The red curve in Fig. 1h is obtained by multiplying the loading vector of sea ice concentration (Fig. 1a) averaged in the boxed area with the amplitude time series (Fig. 1g) according to (1). …]

C7: Page 5, line 9: "the little connectivity" between sea ice reduction and total cloud cover may originate from the fact that sea ice reduction generates two effects that compete against each other: increased latent heat flux tends to increase cloudiness but increasing sensible heat flux tends to reduce it.

R7: We thank the reviewer for enlightening us. As mentioned in our response to your major comments, we included the total cloud pattern since several authors mention the possible role of clouds for Arctic amplification. See our response to Major Comment #1 above.

C8: Page 6, lines 23-25: These are interesting numbers. Please, confirm if these are winter means in 1979-2016 averaged over sea areas north of 60N. It might be interesting to compare them against results of Lupkes et al. (2008), which show air temperature responses to 1% reduction in sea ice cover in different conditions.

R8: It is the result based on the average over the region of sea ice reduction (21°-79.5°E × 75°-79.5°N) in the Barents-Kara Seas; it shows the values of anomalous radiation and surface fluxes for an average sea ice reduction of ~15% (see Fig. 2). Lüpkes et al. (2008) conducted experiments in different settings using a 1D atmospheric model coupled with snow/sea ice model. Therefore, a rigorous comparison is impossible. Our numbers are smaller than those in Lüpkes et al. but are of the same order of magnitude. It is difficult to explain the reasons for this difference, but the absence of horizontal advection is a plausible cause. In the presence of horizontal advection, anomalous temperature and fluxes over sea ice leads are quickly diffused, resulting in reduced local maxima. We can see a hint of horizontal advection in Fig. 1 in the manuscript; while turbulent heat flux is nearly confined to the area of sea ice loss, 2 m air temperature and other variables are smoothed out over a much wider area. We would rather not include this discussion in the revised text, since we eliminated 2 m air temperature increase in the feedback loop. Moreover, our explanation above is somewhat premature and conjectural in nature. [no modification]

C9: Figure 2. Is plot (a) needed at all, as the same line appears in plot (b)?

R9: We removed Figure 2a. [new Figure 2]

5  C10: Figure 4. Add information on colour scales and absolute values. Only the contour intervals are given.

R10: We revised Figure 4 in order to provide the necessary information the reviewer asked. [new Figure 4; figure caption: The red contour is drawn at the value of the contour interval.]

10  C11: Figure 5. Does the shading represent sea ice concentration in all four plots? If yes, why it includes small differences between the plots?

R11: We accidentally used different shading interval for the sea ice concentration field in Fig. 5c. We corrected the figure. [new Figure 5]

C12: Figure 6. Explain better how the time series in days should be interpreted. It cannot be the mean over 1979-2016. Is it from some selected year?

R12: It is the plot of regressed loading vector $B_1(r,t)$ of the sea ice loss mode averaged over the region of sea ice loss (21°-
20  79.5°E × 75°-79.5°N) for different variables. The daily time series are interpreted as typical winter variation of surface fluxes and radiation associated with the sea ice reduction in Fig. 2. Actual data associated with the sea ice loss mode is obtained by multiplying the loading vector with the corresponding PC time series, i.e., the space-time evolution pattern associated with the sea ice loss mode is $T^{(1)}(r,t) = B_1(r,t)T_1(t)$. Thus, the typical time series of surface fluxes and radiation depicted in Fig. 6 are amplifying according to Fig. 1g. We added more description on this figure. [P5 L28-30:
25  Figure 6 shows the winter daily variations of the regressed loading vectors in (6) (terms in curly braces) averaged over the region of sea ice reduction (21°-79.5° E × 75°-79.5° N); it may be interpreted as the atmospheric response to the sea ice reduction shown in Fig. 2.]

C13: Figure 8. Referring to my previous comments, I suggest dropping "2m T increase (0.24)" from the figure, and drawing
30  an arrow directly from increased LW-down to sea ice reduction. Otherwise, provide a good explanation on the causality of the link.

R13: We followed the suggestion of the reviewer. [new Figure 8]

Once CSEOF analysis on the "target" variable is completed as in (1), physically consistent loading vectors of another variable, called the "predictor" variable, are obtained as follows:

Jinju Kim 7/7/2017 9:50 AM

Step 1: $P(r,t) = \sum_n C_n(r,t)P_n(t)$      (CSEOF analysis on a new variable)      (2)

Step 2: $T_n(t) = \sum_{m=1}^{M} \alpha_m^{(n)} P_m(t)$      (regression on PC time series)      (3)

Step 3: $Z_n(r,t) = \sum_{m=1}^{M} \alpha_m^{(n)} C_m(r,t)$      (regressed loading vector)      (4)

Then, the target and predictor variables can be written as

$$\{T(r,t), P(r,t)\} = \sum_n \{B_n(r,t), Z_n(r,t)\} T_n(t).$$      (5)

Namely, the loading vectors of the two variables, $B_n(r,t)$ and $Z_n(r,t)$, share an identical PC time series, $T_n(t)$, for each mode. As a result, the evolution of a physical process manifested as $B_n(r,t)$ and $Z_n(r,t)$ in two different variables is governed by a single amplitude time series. Otherwise, $B_n(r,t)$ and $Z_n(r,t)$ do not represent the same physical process and henceforth are not physically consistent. This process can be repeated for other predictor variables. As a result of regression, then, entire data can be written in the form

$$Data(r,t) = \sum_n \{B_n(r,t), Z_n(r,t), U_n(r,t), \dots\} T_n(t) ,$$      (6)

where the terms in curly braces denote physically consistent evolutions derived from various physical variables. A rigorous mathematical explanation of the regression analysis in CSEOF space can be found in Kim et al. (2015).

Aside from the winter seasonal cycle, the first CSEOF mode derived from the daily winter sea ice concentration data in the Arctic depicts sea ice loss and associated Arctic warming in the Barents and Kara Seas. This mode explains 24% of the total variability of the sea ice concentration in the Artic Ocean and is the focus of investigation in the present study.

**3 Results and Discussion**

Figure 1 shows the winter-averaged pattern of $B_1(r,t)$ together with the regressed patterns from other variables (the terms in the curly braces in (6)). We refer to it as the sea ice loss mode, since the loading vector (Fig. 1a; see also Fig. 2) and the amplitude time series (Fig. 1g) describes the sea ice reduction, together with natural variability of sea ice concentration, in the Barents and Kara Seas during the past 37 years (Fig. 1h). The pattern of sea ice reduction (Fig. 1a) is nearly identical with the trend pattern of sea ice concentration in the Arctic Ocean (see Fig. S1 in the supplementary information). As can be seen in Fig. 1h, the sea ice reduction trend in the Barents and Kara Seas (boxed area in Fig. 1a) is faithfully captured by this mode. In particular, the rate of sea ice loss has significantly increased since 2004-2005 (Vihma, 2014). In association with the sea ice reduction, 2 m air temperature, 850 hPa temperature, specific humidity, upward longwave radiation, downward longwave radiation, and upward heat flux have increased significantly over the region of major sea ice reduction (21°-79.5° E × 75°-79.5° N) (black boxed area in Fig. 1a). As can be seen in Figs. 1a, 1c and 1e, the central areas of anomalous 2 m air temperature, upward longwave radiation and turbulent (sensible + latent) heat flux match well with the region of sea ice loss (Screen and Simmonds, 2010b). On the other hand, the centers of downward longwave radiation and specific humidity match well with that of the 850 hPa air temperature (Figs. 1b, 1d, and 1f).

Sea ice concentration varies slightly on a daily basis, and its fluctuation is less than 2% from the mean value of −14.7% throughout the winter (Fig. 2). In accordance with the reduced sea ice concentration, upward longwave radiation

flux is increased from the warmer sea surface exposed to air. Multiplying the amplitude (PC) time series (Fig. 1g) with the loading vector (Fig. 2) of the sea ice loss mode as in (1), actual sea ice concentration time series is obtained as in Fig. 1h. According to Fig. 1h, sea ice concentration has decreased by ~40% during the last 37 years.

5         Figure 3 shows the anomalous surface (2 m) air temperature, the lower tropospheric geopotential height and wind and the vertical section of anomalous temperature, geopotential height and wind along 60°E and 80°N associated with sea ice reduction. A significant warming is seen in the lower troposphere (e.g., Serreze and Francis, 2006; Serreze et al., 2007; Screen et al., 2013). Note that the anomalous temperature pattern is similar to the second EOF pattern in Graversen et al. (2008). The anomalous temperature and geopotential height are consistent according to the hydrostatic equation (see Fig. S2). Anomalous wind and geopotential height are consistent according to the thermal wind equation. As can be seen, an

10 anticyclonic circulation is established over the region of sea ice loss. This anticyclonic circulation results in advection of warmer air over the Barents and Kara Seas and advection of colder air over the mid-latitude East Asia (Kim and Son, 2016).

        The winter-averaged patterns of anomalous downward longwave radiation and specific humidity look fairly similar to that of 850 hPa air temperature (Figs. 4a and 4b). It appears that the increased downward longwave radiation is the result of the tropospheric warming (Fig. 3). Specific humidity also increases with the tropospheric warming. Note specifically that

15 these changes are observed over or close to the region of sea ice reduction. The patterns of total cloud liquid water and total cloud ice water, which are the key variables for the formation of clouds, also exhibit a strong response over the region of sea ice reduction although their centers of action are shifted toward the Greenland Sea (Fig. 4d). The pattern of total cloud cover, however, does not show any strong cloud activity over the region of sea ice reduction (Fig. S3 in the supplementary information); it should be understood that cloud cover is a difficult variable to simulate accurately in a reanalysis model.

20 Therefore, we postulate that the increased downward longwave radiation is due to the increased 850 hPa air temperature and the greenhouse effect produced by the increased specific humidity. Further note that net (upward minus downward) longwave radiation is positive over the region of major sea ice reduction, whereas it is slightly negative over the surrounding areas (Fig. 4c). Thus, at the surface level, there is a net loss of longwave energy over the region of sea ice reduction, while there is a net gain of longwave radiation over the surrounding area.

25         A prominent source of energy available for heating the atmospheric column is the increased turbulent heat flux from the sea surface exposed to air due to sea ice reduction (Fig. 5). Figure 6 shows the winter daily variations of the regressed loading vectors in (6) (terms in curly braces) averaged over the region of sea ice reduction (21°-79.5° E × 75°-79.5° N); it may be interpreted as atmospheric response to the sea ice reduction shown in Fig. 2. Although the total (area-weighted) magnitudes of sensible and latent heat fluxes are generally smaller than those of upward and downward longwave radiation

[revised manuscript text omitted]

of 2025). A linear fit, the most conservative of the three but with the largest residual error, predicts a complete disappearance of sea ice in this area by 2065 (see Fig. S4).

It should be pointed out that this feedback process could develop in other areas of the Arctic Ocean. If sea ice refreezing is delayed in late fall/winter, increased turbulent heat flux from the open sea surface will make it more difficult for

5    sea surface to refreeze, ultimately leading to the feedback process in Fig. 8. It is, of course, difficult to determine when this should occur, since environmental factors differ from one location to another. Finally, it should be mentioned that the feedback process does not seem to be sensitive to a choice of the dataset. A similar experiment conducted by using the NCEP reanalysis data produces essentially identical results except for a slight overestimation of the strength of the anomalous patterns in Fig. 1a-f and Fig. 6 (see Figs. S5 and S6 in the supplementary information).

[revised manuscript text omitted]
 red curve in Fig. 1h is obtained by multiplying the loading vector of sea ice concentration (Fig. 1a) averaged in the boxed area with the amplitude time series (Fig. 1g) according to (1). The green contours in (b)-(f) represent sea ice concentration in (a). The numbers in parenthesis are contour intervals and negative contours are dashed.

**Figure 2.** Anomalous daily sea ice concentration (blue) and upward longwave radiation averaged over the region of sea ice loss (21°-79.5° E × 75°-79.5° N) with respective mean values (straight lines). Winter days are counted from December 1.

**Figure 3**. Winter-averaged patterns of anomalous atmospheric condition: (a) 2m air temperature, (b) lower tropospheric (1000-900 hPa) geopotential height and wind, (c) vertical cross section along 60° E of lower tropospheric (1000-850 hPa) air temperature, geopotential height and wind, and (d) along 80°N. Contour intervals are in parenthesis in (a) and (b). Temperature is in shading (0.4 K), geopotential height is in black contours (3 m), and (c) zonal and (d) meridional winds are in blue contours (0.2 m s$^{-1}$).

**Figure 4**. Winter-averaged patterns of (a) 850 hPa air temperature (shading) and 2 m air temperature (contour), (b) 900-hPa specific humidity (shade) and downward longwave radiation at surface (contour), (c) net (upward minus downward) longwave radiation at surface (shade) and SAT (contour), and (d) total cloud liquid water (shade) and total cloud ice water (contour) for the sea ice loss mode. The red contour is drawn at the value of the contour interval. The green contours in (a)-(d) represent the reduction of sea ice concentration.

**Figure 5**. Winter average pattern of sea ice loss mode in the Barents and Kara Seas: (a) sea ice (%, shading), 2 m air temperature (red contour) and 850 hPa temperature (black contour), (b) upward longwave radiation (red contour) and downward longwave radiation (black contour), (c) sensible heat flux (red contour) and latent heat flux (black contour), and (d) net energy balance (sensible heat flux + latent heat flux + upward longwave radiation – downward longwave radiation).

**Figure 6**. Daily patterns of variability over the region of sea ice loss (21°-79.5° E × 75°-79.5° N): (a) upward longwave radiation (blue dashed), downward longwave radiation (blue dotted), net longwave radiation (blue solid) with its mean value (blue straight line), sensible heat flux (red dashed), latent heat flux (red dotted), and turbulent heat flux (red solid) with its mean value (red straight line), (b) 2 m air temperature (red), 850 hPa air temperature × 2 (black), and upward longwave radiation (blue), and (c) same as (b) except for the regressed downward longwave radiation (blue). The straight lines in (b) and (c) represent the winter mean value of anomalous 2 m air temperature. Correlation of upward and downward longwave radiations with 2 m air temperature is respectively 0.88 and 0.91, whereas with 850 hPa air temperature is 0.66 and 0.85.

Jinju Kim 7/7/2017 10:25 AM

Jinju Kim 7/7/2017 10:01 AM

Jinju Kim 7/7/2017 10:02 AM

Jinju Kim 7/7/2017 10:02 AM

Jinju Kim 7/7/2017 10:24 AM

**Figure 7.** Lagged correlations: (a) correlation of upward (solid lines) and downward (dotted lines) longwave radiations with 2 m air temperature (blue), 850 hPa temperature (red), and sea ice concentration (black), and (b) a blow-up of the boxed region in (a). Longwave radiation lags the other variable for a positive lag. Lagged correlation between 2 m air temperature and 850 hPa air temperature (black dashed line); 2 m air temperature leads 850 hPa temperature for a positive lag.

5  **Figure 8**. A proposed mechanism of polar amplification. Increased net upward energy flux increases air temperature. As a result, downward longwave radiation increases, which results in sea ice melting. This loop seems to amplify by ~8.9% annually.

Jinju Kim 7/7/2017 10:03 AM

(a) SIC  (2 %)  &  2m AIR T  (0.5° C)

(b) 1000-850 hPa SH  (3×10⁻² g Kg⁻¹)

(c) ULW at SFC  (2 W m⁻²)

(d) DLW at SFC  (2 W m⁻²)

(e) TURBULENT FLUX  (4 W m⁻²)

(f) 850 hPa T  (0.2° C)

[Figure]

Jinju Kim 7/7/2017 10:07 AM

(a) SIC  (2 %)  &  2m AIR T  (0.5° C)

(c) ULW at SFC  (2 W m⁻²)

(e) TURBULENT FLUX  (4 W m⁻²)

[Figure]

**Figure 1.** Winter (Dec. 1-Feb. 28) average patterns of sea ice loss mode: (a) sea ice (shading) and 2 m air temperature (contour), (b) 1000-850 hPa specific humidity, (c) upward longwave radiation, (d) downward longwave radiation, (e) turbulent (sensible + latent) heat flux, (f) 850 hPa air temperature, (g) the corresponding amplitude change (red solid curve) and the amplification curve (blue dashed curve), and (h) actual sea ice change in the sea-ice loss region (21°–79.5° E × 75°–79.5° N; the boxed area in (a)) of the Barents and Kara Seas (black dotted curve; extended until 2017 based on new data), sea ice change according to the sea ice loss mode (red curve), projection based on the amplification curve (blue dashed curve). The red curve in Fig. 1h is obtained by multiplying the loading vector of sea ice concentration (Fig. 1a) averaged in the boxed area with the amplitude time series (Fig. 1g) according to (1). The green contours in (b)-(f) represent sea ice concentration in (a). The numbers in parenthesis are contour intervals and negative contours are dashed.

Jinju Kim 7/7/2017 10:25 AM

[Figure]

**Figure 2.** Anomalous daily sea ice concentration (blue) and upward longwave radiation averaged over the region of sea ice loss (21°-79.5° E × 75°-79.5° N) with respective mean values (straight lines). Winter days are counted from December 1.

[Figure]

Jinju Kim 7/7/2017 10:10 AM

[Figure]

[Figure]

**Figure 3.** Winter-averaged patterns of anomalous atmospheric condition: (a) 2 m air temperature, (b) lower tropospheric (1000-900 hPa) geopotential height and wind, (c) vertical cross section along 60° E of lower tropospheric (1000-850 hPa) air temperature, geopotential height and wind, and (d) along 80° N. Contour intervals are in parenthesis in (a) and (b). Temperature is in shading (0.4 K), geopotential height is in black contours (3 m), and (c) zonal and (d) meridional winds are in blue contours (0.2 m s⁻¹).

[Figure]

[Figure]

**Figure 4**. Winter-averaged patterns of (a) 850 hPa air temperature (shading) and 2 m air temperature (contour), (b) 900-hPa specific humidity (shade) and downward longwave radiation at surface (contour), (c) net (upward minus downward) longwave radiation at surface (shade) and SAT (contour), and (d) total cloud liquid water (shade) and total cloud ice water (contour) for the sea ice loss mode. The red contour is drawn at the value of the contour interval. The green contours in (a)-(d) represent the reduction of sea ice concentration.

Jinju Kim 7/7/2017 10:11 AM

Jinju Kim 7/7/2017 10:12 AM

Jinju Kim 7/7/2017 10:26 AM

[Figure]

[Figure]

Figure 5. Winter average pattern of sea ice loss mode in the Barents and Kara Seas: (a) sea ice (%, shading), 2 m air temperature (red contour) and 850 hPa temperature (black contour), (b) upward longwave radiation (red contour) and downward longwave radiation (black contour), (c) sensible heat flux (red contour) and latent heat flux (black contour), and (d) net energy balance (sensible heat flux + latent heat flux + upward longwave radiation − downward longwave radiation).

[Figure]

**Figure 6.** Daily patterns of variability over the region of sea ice loss (21°-79.5° E × 75°-79.5° N): (a) upward longwave radiation (blue dashed), downward longwave radiation (blue dotted), net longwave radiation (blue solid) with its mean value (blue straight line), sensible heat flux (red dashed), latent heat flux (red dotted), and turbulent heat flux (red solid) with its mean value (red straight line), (b) 2 m air temperature (red), 850 hPa air temperature × 2 (black), and upward longwave radiation (blue), and (c) same as (b) except for the regressed downward longwave radiation (blue). The straight lines in (b) and (c) represent the winter mean value of anomalous 2 m air temperature. Correlation of upward and downward longwave radiations with 2 m air temperature is respectively 0.88 and 0.91, whereas with 850 hPa air temperature is 0.66 and 0.85.

[Figure]

[Figure]

Jinju Kim 7/7/2017 10:14 AM

**Figure 7**. Lagged correlations: (a) correlation of upward (solid lines) and downward (dotted lines) longwave radiations with 2 m air temperature (blue), 850 hPa temperature (red), and sea ice concentration (black), and (b) a blowup of the boxed region in (a). Longwave radiation lags the other variable for a positive lag. Lagged correlation between 2 m air temperature and 850 hPa air temperature (black dashed line); 2 m air temperature leads 850 hPa temperature for a positive lag.

[Figure]

**Figure 8.** A proposed mechanism of polar amplification. Increased net upward energy flux increases air temperature. As a result, downward longwave radiation increases, which results and sea ice melting. This loop seems to amplify by ~8.9 % annually.

**Supplementary Information**

**Understanding the Mechanism of Arctic Amplification and Sea Ice Loss**

Kwang-Yul Kim[1], Jinju Kim[1], Saerim Yeo[2], Hanna Na[3], Benjamin D. Hamlington[4], and Robert R. Leben[5]

[1]School of Earth and Environmental Sciences, Seoul National University, 1 Gwanak-ro, Gwanak-gu, Seoul 08826, Republic of Korea
[2]APEC Climate Center 1463, Haeundae-gu, Busan 48058, Republic of Korea
[3]Ocean Circulation and Climate Research Center, Korea Institute of Ocean Science and Technology, Ansan, 15627, Republic of Korea
[4]Department of Ocean, Earth and Atmospheric Sciences, 4600 Elkhorn Avenue, Room 406, Old Dominion University, Norfolk, Virginia 23529, USA
[5]Colorado Center for Astrodynamics Research, Department of Aerospace Engineering Sciences, ECNT 320, 431 UCB, University of Colorado, Boulder, Colorado 80309-0431, USA

*Correspondence to*: Kwang-Yul Kim (kwang56@snu.ac.kr)

[Figure]

**Figure S1**. (a) The yearly trend (%) of sea ice reduction in the Arctic Ocean during 1979-2016, (b) the winter averaged loading vector of the sea ice loss mode, (c) the corresponding PC (amplitude) time series, and (d) actual sea ice concentration in the boxed area (black curve), sea ice concentration according to the sea ice loss mode (red curve) and a projection (red dashed curve) based on the exponential fit of the amplitude time series in (c).

Figure S1a is obtained based on a linear trend of sea ice concentration at each grid point based on the ERA-Interim sea ice concentration from 1979-2016. This pattern is nearly identical with the sea ice loss mode discussed in the main text (Fig. S1b). Sea ice reduction is most conspicuous in the Barents-Kara Seas. The amount of sea ice reduction based on the sea ice loss mode is obtained by multiplying the loading vector (Fig. S1b) with its amplitude time series (Fig. S1c), resulting in Fig. S1d. As can be seen in Fig. S1d, the sea ice concentration change due to the sea ice loss mode (red curve) is similar to the actual data (black curve) with a fairly similar rate of trend.

[Figure]

**Figure S2.** The pressure layer thickness ($\Delta Z = Z(p_1) - Z(p_0)$) derived from the geopotential height pattern in Fig. 3 in the text and that derived from the hydrostatic equation (contour). The red contour represents the thickness of 1.5 m. The level

5    $p_1$ is the level used for plotting and $p_0$ is the pressure level below $p_1$ at the interval of 25 hPa.

The shaded geopotential height anomaly in this figure is obtained directly from the geopotential height field in Fig. 3 in the main text, i.e.,

10    $$(dZ)_j = Z_j - Z_{j-1}, \tag{1}$$

where $j$ is an index for the vertical level. The contoured geopotential height anomaly is obtained from the temperature field in Fig. 3 in the text, i.e.,

$$(dZ)_j = -\frac{R\langle T\rangle_j}{g}(d\ln p)_j, \tag{2}$$

where

15    $$(dZ)_j = Z_j - Z_{j-1}, \quad \langle T\rangle_j = (T_j + T_{j-1})/2, \quad (d\ln p)_j = \ln p_j - \ln p_{j-1}. \tag{3}$$

As can be seen in the figure, the anomalous geopotential height field is nearly in hydrostatic balance with the anomalous temperature field. The difference is partially due the use of layer mean temperature $\langle T\rangle$ in a finite-difference approximation of the hydrostatic equation in (2). Thus, it seems that the release of energy in the form of radiation and heat flux changes the temperature, and geopotential height in the lower troposphere adjusts in accordance with the hydrostatic balance.

[Figure]

(a) 850 hPa T (0.2° C) & SAT (0.5° C)  (b) TCC (0.5 %)

(c) DLW & ULW at SFC (2 W m⁻²)  (d) SH (0.02 g kg⁻¹) & DLW (2 W m⁻²)

**Figure S3**. The DJF patterns of 850 hPa air temperature (shading) and 2 m air temperature (contour) (a), total cloud cover (b), downward (shade) and upward (contour) longwave radiation at surface (c), and 900-hPa specific humidity (shade) and downward longwave radiation at surface (contour) (d) for the sea ice loss mode. The green and purple contours in (a)-(d) represent the reduction of sea ice concentration.

This figure shows the winter (DJF) averaged patterns of several key variables associated with the sea ice reduction. As can be seen, anomalous patterns of all the variables exhibit strong coherence with that of sea ice reduction except for total cloud cover. The pattern of total cloud cover associated with the sea ice loss mode does not exhibit any strong cloud activity over the region of sea ice reduction, suggesting little connectivity between sea ice reduction and change in cloud cover. However, total cloud liquid water and total cloud ice water, which are two important elements for the production of clouds are reasonably consistent with the pattern of sea ice reduction (see Fig. 4 in the main text).

[Figure]

**Figure S4**. Actual sea ice change in the sea-ice loss region (21°–79.5°E, 75°–79.5 °N) of the Barents and Kara Seas (black dotted curve; updated until Feb. 2017 using new dataset), sea ice change according to the sea ice loss mode (red curve), projections based on the exponential fitting (blue dashed curve), quadratic fitting (dash-dot curve), and linear fitting (dotted curve) of the PC time series.

This figure shows projections of sea ice concentration in the sea-ice loss region of the Barent-Kara Seas based on a linear fit (dotted curve), a quadratic fit (dash-dot curve), and an exponential fit (dashed). Residual variance is measured by

$$\varepsilon^2 = var\big(S(t) - F(t)\big),$$

where $S(t)$ is the sea ice concentration curve (black curve in Fig. S4) and $F(t)$ is a fit. The exponential fit results in the least residual variance, whereas the linear fit the largest residual variance. The residual variance of the quadratic fit is similar to that of the exponential fit. Sea ice in the region (21°–79.5°E, 75°–79.5 °N) disappears completely by 2025 (2030, 2065) according to the exponential (quadratic, linear) fit. According to the newly available data, sea ice concentration in this area is the lowest during 2016 winter (Dec. 2016-Feb. 2017; see the dotted line in Fig. S4).

[Figure]

**Figure S5**. Winter (Dec. 1-Feb. 28) average patterns of sea ice loss mode derived from the NCEP reanalysis data: (a) sea ice (shading) and 2 m air temperature (contour), (b) 1000-850 hPa specific humidity, (c) upward longwave radiation, (d) downward longwave radiation, (e) turbulent (sensible + latent) heat flux, (f) 850 hPa air temperature.

This figure shows the winter-averaged patterns of key variables from the NCEP reanalysis product (1979-2016) associated with the sea ice loss mode. The target variable is the ERA-Interim sea ice concentration as in the main text. This figure is fairly similar to Fig. 1a-f in the text except for a small difference in magnitude. Thus, an essentially identical physical process is identified in the NCEP reanalysis product. This result indicates that the physical mechanism addressed in the present study is not overly sensitive to the choice of the dataset for analysis.

[Figure]

**Figure S6**. Daily pattern of variability over the region of sea ice loss (21°-79.5°E × 75°-79.5°N) derived from the NCEP reanalysis data: (a) 2 m air temperature (red), 850 hPa air temperature × 2 (black), and upward longwave radiation (blue), and (b) same as (a) except for the regressed downward longwave radiation (blue). The straight line represents the winter mean value of anomalous 2 m air temperature. Correlation of upward and downward longwave radiation with 2 m air temperature is respectively 0.95 and 0.94, whereas correlation with 850 hPa air temperature is respectively 0.81 and 0.86.

This figure shows the daily evolution of surface (2 m) air temperature, 850 hPa air temperature, upward longwave radiation and downward longwave radiation during winter in response to sea ice reduction in the Barents-Kara Seas as in Fig. 2 in the main text. This result is obtained by using the NCEP reanalysis data. A comparison with Fig. 6 in the main text shows that the response of atmospheric variables to the sea ice reduction in the Barents-Kara Seas as identified from the NCEP reanalysis product is fairly similar to that derived from the ERA-Interim reanalysis product. This figure together with Fig. S4 indicates that the physical process of sea ice reduction and Arctic warming discussed in the text is not sensitive to the choice of analysis dataset.

---

## Author Comment (AC2) · 7 Jul 2017

Comment1(C1): The paper applies a sort of regression analysis to the wintertime (JF) sea ice loss in the Barents-Kara seas. The review of prior literature on Arctic amplification and sea ice loss is often confusing, including in the definition of key concepts such as Arctic amplification or albedo feedback.

Response1(R1): Reviewer #2's comments are difficult to address because of the lack of detail in the review. More specificity is needed so that we can address the concerns of the reviewer. Which parts of the manuscript are confusing in terms of Arctic amplification and sea ice loss? What about the definition of Arctic amplification and albedo feedback is confusing? Arctic amplification represents a rapid warming of the Arctic

temperature, the physical interpretation of which may vary from one group of scientists to another. Albedo feedback is a feedback produced by albedo change. In summer, sea ice reduction decreases surface albedo in the Arctic Ocean, thereby increasing the absorption of solar energy in the ocean. This is referred to as albedo feedback in the manuscript.

C2: Based on the explanations given in the manuscript, I cannot understand the authors' methodology sufficiently to judge its value. The manuscript lacks a critical appreciation of the method, e.g. a discussion of how much of the time series and trend is actually captured by the first 'mode' obtained in the analysis.

R2: The methodology was published 20 years ago and has been used in many papers. We cannot repeat the full discussion on the methodology every time a paper is submitted. That is why three key references on the methodology have been added. We tried to improve the method section by including more specific details. [P3 L18-23: … its amplitude varies from one year to another according to the corresponding PC time series. CSEOF loading vectors are mutually orthogonal to each other in space and time and represent distinct physical processes. The principal component (PC) time series, $T_n(t)$ are uncorrelated with (and are often nearly independent of) each other. Thus, each loading vector depicts a temporal evolution of spatial patterns seen in a physical process (such as El Niño or seasonal cycle), and corresponding PC time series describes a long-term modulation of the amplitude of the physical process.] [P4 L14-15: A rigorous mathematical explanation of the regression analysis in CSEOF space can be found in Kim et al. (2015).]

We also added how much of the total variability is explained by the sea ice loss mode. As can be seen in Fig. R1, the trend of sea ice reduction is most conspicuous in the Barents-Kara Seas. Figure R1a is very similar to Fig. 1a in the manuscript. Figure R1b also shows that sea ice reduction in the Barents-Kara Seas (red-boxed area in Fig. R1a) is well explained by the sea ice loss mode (red curve).

We made the following change in the revised manuscript: [P4 L17-18: Aside from the winter seasonal cycle, the first CSEOF mode derived from the daily winter sea ice concentration data depicts sea ice loss and associated Arctic warming in the Barents and Kara Sea. This mode explains 24% of the total variability of the sea ice concentration in the Artic Ocean and is the focus of investigation in the present study.] [P4 L23-26... 37 years (Fig. 1h). The pattern of sea ice reduction (Fig. 1a) is nearly identical with the trend pattern of sea ice concentration in the Arctic Ocean (see Fig. S1 in the supplementary information). As can be seen in Fig. 1h, the sea ice reduction trend in the Barents and Kara Seas (boxed area in Fig. 1a) is faithfully captured by this mode.]

C3: My fundamental concern with the manuscript is that it uses correlations to establish causalities and feedbacks, with little regard to the physical and meteorological phenomena discussed. As an example, the feedback loop suggested as a key result of the paper begins with sea ice reduction which supposedly causes warming of 850 hPa temperatures. The alternative explanation that warm air advection contributes to sea ice loss is at least as plausible, but not even mentioned in the manuscript.

R3: As we discussed in the "method of analysis" section, the CSEOF technique writes the space-time data in the form

$$T(r,t) = \Sigma_n B_n(r,t)T_n(t), \qquad (1)$$

where $B_n(r,t)$ represents the deterministic (physical) evolution associated with the nth CSEOF mode and $T_n(t)$ is the corresponding stochastic amplitude time series. Unlike EOF loading vectors, each CSEOF loading vector is time dependent and depicts physical (deterministic) evolution. In order to obtain physically consistent loading vectors from different variables, we used regression analysis in CSEOF space, the procedure of which is delineated in the manuscript. After regression analysis in CSEOF space, the entire dataset can be written as

$$Data(r,t) = \Sigma_n\{B_n(r,t), Z_n(r,t), U_n(r,t), \cdots\}T_n(t), \qquad (2)$$

where the terms in curly braces represent loading vectors from different variables. They are consistent in a physical sense.

Our statements are not solely based on correlations. At the very outset, we stated clearly that we would make a quantitative estimate of individual processes involved in an accelerated loss of sea ice concentration (P1 L22, P2 L28, P2 L30). It is the set of loading vectors in (2) that we are concerned with. For example, Fig. R2 above shows the time-averaged patterns of $\{B_n(r,t), Z_n(r,t), U_n(r,t), \cdots\}$ for the first CSEOF mode (sea ice loss mode). It shows how each variable behaves in accordance with the sea ice reduction in the Barents-Kara Seas. Another example is Fig. R3 above, where daily variation of each variable averaged over the region of sea ice reduction (red box in Fig. R1a). Based on this figure, we can understand how physical variables respond to the sea ice reduction over the Barents-Kara Seas, and in what way two or more variables are physically related with each other. As can be seen in Fig. R3, several variables evolve in a very similar manner over the region of sea ice reduction. It also shows how much the winter mean of each variable changes due to sea ice reduction. We do not know how correlation analysis could be used to make the physical inferences similar to those found in the present study.

As the reviewer mentioned, there are other processes such as warm advection that may be important for Arctic amplification and sea ice reduction. As can be seen in Fig. R4, there is a net convergence of moisture transport and heat transport over a region of sea ice reduction, although the center of action is over the Greenland Sea. Thus, moisture and heat transport from lower latitudes apparently affects the variation of sea ice concentration. Figure R5 further shows that there is appreciable correlation between the variation of specific humidity and convergence of moisture transport (corr=0.62) and between the variation of lower tropospheric temperature and convergence of heat transport (corr=0.33). Thus, it seems that both the convergence of moisture transport and the convergence of heat transport are responsible for the variation of specific humidity and temperature in the lower troposphere. On the other hand, the convergence

of horizontal transport of moisture cannot explain one essential element of the specific humidity anomaly–the mean of the anomalous specific humidity. As can be seen in Figure R5a, the mean of moisture convergence is close to $0.6 \times 10^{-6}$ g/kg/sec, which amounts to $\sim 0.05$ g/kg of moisture. This value explains only about 17% of the mean value of anomalous specific humidity ($\sim 0.3$ g/kg); the remainder should derive from a vertical process.

Consider the following moisture conservation equation:

$$\frac{\partial q}{\partial t} = -\vec{u} \cdot \nabla q + S \doteq -\nabla \cdot (q\vec{u}) + S = -\nabla_h \cdot (q\vec{u}) - \frac{\partial (qw)}{\partial z} + S. \qquad (3)$$

According to Fig. R5, the convergence of the horizontal moisture transport is not so effective as the convergence of the vertical moisture transport in the equation above in terms of increasing the mean of specific humidity. A positive convergence is offset by a negative convergence and vice versa, resulting in a small net increase in the mean of specific humidity. As can be seen in Fig. R6, the anomalous evaporation due to sea ice reduction is positive throughout the winter and its magnitude is reasonable in comparison with the increase in specific humidity. The two time series in Fig. R6 are negatively correlated (except for the mean), indicating that increased (decreased) specific humidity due to positive (negative) convergence of moisture transport reduces (augments) evaporation from the surface of the ocean; this is a reasonable explanation according to the bulk formula.

Likewise, the variation of the thermal advection and the subsequent convergence of the heat flux are highly correlated with the variation of downward longwave radiation and the lower tropospheric (850 hPa) temperature (see Fig. R5b). On the other hand, the small mean value of the convergence of the horizontal heat flux cannot explain the significant nonzero mean of the anomalous downward longwave radiation or the anomalous lower tropospheric (850 hPa) temperature. Thus, we conclude that the vertical process should be invoked in order to account for the significant changes in the

means of the variables over the Barents-Kara Seas. We did not simply ignore the contributions of moisture transport and heat transport from lower latitudes. Rather, this is a serious issue and requires more detailed calculation and convincing demonstration, which we considered beyond the scope of the present paper. We, however, acknowledge that we restricted ourselves to processes acting in the Arctic, ignoring the forcing from lower latitudes. Based on this discussion we made the following changes: [P3 L4-5: It should be noted that our discussion is restricted to processes in the Arctic; forcing from lower latitudes can also be important in the process of Arctic amplification and sea ice reduction.]

C4: I further do not see any justification for fitting an exponential to the time series of sea ice loss in the Barents-Kara seas, and far less for using that fit to make a prediction on when this ocean area would remain ice-free in winter.

R4: As can be seen in Fig. 1h in the manuscript, the exponential fitting looks reasonable in describing the change in sea ice concentration in the Barents-Kara Seas. One can use a linear or quadratic fit to make a similar prediction (see Fig. R7). Predictions are predictions whether it is based on the exponential fitting or the fitting of a low order polynomial; uncertainty is inherent in a prediction. We added the predictions based on linear fit and quadratic fit as supplementary information for the benefit of the readers. The reason why we chose the exponential fit (not on the sea ice concentration but on the PC time series of the sea ice loss mode in Fig. 1g in the manuscript) is that it yields the least residual error. The residual error based on a quadratic fit is similar to that of the exponential fit whereas a linear fit yields the largest residual variance among the three. We included some of this discussion and Fig. R7 as supplementary information. We also modified the text as follows: [P1 L24: ... sea ice will completely disappear in the Barents and Kara Seas by as early as 2025, although a conservative linear fit delays it until 2065.] [P7 L31: We fitted an exponential curve to the amplitude time series of the sea ice loss mode (Fig. 1g); an exponential fitting is chosen, since it minimizes the residual error. Our calculation shows that sea ice in the sea-ice loss region (21°-

79.5°E × 75°-79.5°N) of the Barents and Kara Seas may completely melt by as early as 2025 (Fig. 1h) unless impeded by other naturally occurring variability. A quadratic fit results in a similar result (2030 instead of 2025). A linear fit, the most conservative of the three but with the largest residual error, predicts a complete disappearance of sea ice in this area by 2065 (see Fig. S4).] We also updated the sea ice concentration curve using the 2017 sea ice data (see new Fig. 1h).

C5: In conclusion, I regret to say that the manuscript fails to meet basic scientific standards.

R5: We have addressed all specific comments and are not sure what standards we have failed to meet. We would be happy to address those if they would be identified.

** The combined response file including a marked-up manuscript is attached.

Please also note the supplement to this comment:
https://www.the-cryosphere-discuss.net/tc-2017-39/tc-2017-39-AC2-supplement.pdf

[Figure]

[Figure]

**Fig. 1.** The sea ice trend (% per year) in the Arctic Ocean (left panel) and the sea ice concentration in the red-boxed area of the Barents-Kara Seas (right panel).

(a) 850 hPa T (0.2° C) & SAT (0.5° C)

(b) SH (0.02 g kg⁻¹) & DLW (2 W m⁻²)

(c) NLW at SFC (1 W m⁻²) & SAT (0.5° C)

(d) TCLW (0.5 g kg⁻¹) & TCIW (1 g kg⁻¹)

**Fig. 2.** The winter (DJF) averaged regressed patterns of several physical variables (reproduced from Fig. 4 in the manuscript). The caption of each panel shows the shading (contour) interval for the first (se

[Figure]

**Fig. 3.** The daily patterns of variability over the region of sea ice loss (21°-79.5°E × 75°-79.5°N) (reproduced from Figure 6 in the manuscript).

DJF

DJF

**Fig. 4.** Winter-averaged (left panel) moisture transport (streamline) and its convergence (shade) and (right panel) heat transport (streamline) and its convergence (shade) in the lower troposphere (1000-850 hP

[Figure]

**Fig. 5.** The daily time series of anomalous specific humidity and anomalous moisture conver-
gence averaged over the sea ice loss region (21°-79.5°E × 75°-79.5°N) in the Barents-Kara
Seas. This particular time

**Fig. 6.** The daily variation of specific humidity (red) and evaporation (blue) averaged over the region of sea ice reduction (21°-79.5°E × 75°-79.5°N) in the Barents-Kara Seas.

[Figure]

Fig. 7. Actual sea ice change in the sea-ice loss region (21°-79.5°E × 75°-79.5°N) of the Barents and Kara Seas (black dotted curve), sea ice change according to the sea ice loss mode (red curve), projections

**Supplement:**

**Response to the Comments of Reviewer #2**

Comment1(C1): The paper applies a sort of regression analysis to the wintertime (JF) sea ice loss in the Barents-Kara seas. The review of prior literature on Arctic amplification and sea ice loss is often confusing, including in the definition of key concepts such as Arctic amplification or albedo feedback.

Response1(R1): Reviewer #2's comments are difficult to address because of the lack of detail in the review. More specificity is needed so that we can address the concerns of the reviewer. Which parts of the manuscript are confusing in terms of Arctic amplification and sea ice loss? What about the definition of Arctic amplification and albedo feedback is confusing? Arctic amplification represents a rapid warming of the Arctic temperature, the physical interpretation of which may vary from one group of scientists to another. Albedo feedback is a feedback produced by albedo change. In summer, sea ice reduction decreases surface albedo in the Arctic Ocean, thereby increasing the absorption of solar energy in the ocean. This is referred to as albedo feedback in the manuscript.

C2: Based on the explanations given in the manuscript, I cannot understand the authors' methodology sufficiently to judge its value. The manuscript lacks a critical appreciation of the method, e.g. a discussion of how much of the time series and trend is actually captured by the first 'mode' obtained in the analysis.

R2: The methodology was published 20 years ago and has been used in many papers. We cannot repeat the full discussion on the methodology every time a paper is submitted. That is why three key references on the methodology have been added. We tried to improve the method section by including more specific details. [P3 L18-23: … its amplitude varies from one year to another according to the corresponding PC time series. CSEOF loading vectors are mutually orthogonal to each other in space and time and represent distinct physical processes. The principal component (PC) time series, $T_n(t)$ are uncorrelated with (and are often nearly independent of) each other. Thus, each loading vector depicts a temporal evolution of spatial patterns seen in a physical process (such as El Niño or seasonal cycle), and corresponding PC time series describes a long-term modulation of the amplitude of the physical process.] [P4 L14-15: A rigorous mathematical explanation of the regression analysis in CSEOF space can be found in Kim et al. (2015).]

We also added how much of the total variability is explained by the sea ice loss mode. As can be seen in Fig. R1, the trend of sea ice reduction is most conspicuous in the Barents-Kara Seas. Figure R1a is very similar to Fig. 1a in the manuscript. Figure R1b also shows that sea ice reduction in the Barents-Kara Seas (red-boxed area in Fig. R1a) is well explained by the sea ice loss mode (red curve).

[Figure]

Figure R1. The sea ice trend (% per year) in the Arctic Ocean (left panel) and the sea ice concentration in the red-boxed area of the Barents-Kara Seas (right panel).

We made the following change in the revised manuscript: [P4 L17-18: Aside from the winter seasonal cycle, the first CSEOF mode derived from the daily winter sea ice concentration data depicts sea ice loss and associated Arctic warming in the Barents and Kara Sea. This mode explains 24% of the total variability of the sea ice concentration in the Artic Ocean and is the focus of investigation in the present study.] [P4 L23-26… 37 years (Fig. 1h). The pattern of sea ice reduction (Fig. 1a) is nearly identical with the trend pattern of sea ice concentration in the Arctic Ocean (see Fig. S1 in the supplementary information). As can be seen in Fig. 1h, the sea ice reduction trend in the Barents and Kara Seas (boxed area in Fig. 1a) is faithfully captured by this mode.]

C3: My fundamental concern with the manuscript is that it uses correlations to establish causalities and feedbacks, with little regard to the physical and meteorological phenomena discussed. As an example, the feedback loop suggested as a key result of the paper begins with sea ice reduction which supposedly causes warming of 850 hPa temperatures. The alternative explanation that warm air advection contributes to sea ice loss is at least as plausible, but not even mentioned in the manuscript.

R3: As we discussed in the "method of analysis" section, the CSEOF technique writes the space-time data in the form

$$T(r,t) = \sum_n B_n(r,t) T_n(t), \tag{1}$$

where $B_n(r,t)$ represents the deterministic (physical) evolution associated with the $n$th CSEOF mode and $T_n(t)$ is the corresponding stochastic amplitude time series. Unlike EOF loading vectors, each CSEOF loading vector is time dependent

and depicts physical (deterministic) evolution. In order to obtain physically consistent loading vectors from different variables, we used regression analysis in CSEOF space, the procedure of which is delineated in the manuscript. After regression analysis in CSEOF space, the entire dataset can be written as

$$Data(r,t) = \sum_n \{B_n(r,t), Z_n(r,t), U_n(r,t), \dots\} T_n(t), \qquad (2)$$

5   where the terms in curly braces represent loading vectors from different variables. They are consistent in a physical sense.

[Figure]

Figure R2. The winter (DJF) averaged regressed patterns of several physical variables (reproduced from Fig. 4 in the
10   manuscript). The caption of each panel shows the shading (contour) interval for the first (second) variable. The red contour is at the first contour level (contour value is identical with the contour interval).

[Figure]

Figure R3. The daily patterns of variability over the region of sea ice loss (21°-79.5° E × 75°-79.5° N) (reproduced from Figure 6 in the manuscript).

5   Our statements are not solely based on correlations. At the very outset, we stated clearly that we would make a quantitative estimate of individual processes involved in an accelerated loss of sea ice concentration (P1 L22, P2 L28, P2 L30). It is the set of loading vectors in (2) that we are concerned with. For example, Fig. R2 above shows the time-averaged patterns of $\{B_n(r,t), Z_n(r,t), U_n(r,t), ...\}$ for the first CSEOF mode (sea ice loss mode). It shows how each variable behaves in accordance with the sea ice reduction in the Barents-Kara Seas. Another example is Fig. R3 above, where daily variation of

10   each variable averaged over the region of sea ice reduction (red box in Fig. R1a). Based on this figure, we can understand how physical variables respond to the sea ice reduction over the Barents-Kara Seas, and in what way two or more variables

[Figure]

Figure R4. Winter-averaged (left panel) moisture transport (streamline) and its convergence (shade) and (right panel) heat transport (streamline) and its convergence (shade) in the lower troposphere (1000-850 hPa) associated with the sea ice loss mode.

are physically related with each other. As can be seen in Fig. R3, several variables evolve in a very similar manner over the region of sea ice reduction. It also shows how much the winter mean of each variable changes due to sea ice reduction. We do not know how correlation analysis could be used to make the physical inferences similar to those found in the present study.

As the reviewer mentioned, there are other processes such as warm advection that may be important for Arctic amplification and sea ice reduction. As can be seen in Fig. R4, there is a net convergence of moisture transport and heat transport over a region of sea ice reduction, although the center of action is over the Greenland Sea. Thus, moisture and heat transport from lower latitudes apparently affects the variation of sea ice concentration. Figure R5 further shows that there is appreciable correlation between the variation of specific humidity and convergence of moisture transport (corr=0.62) and between the variation of lower tropospheric temperature and convergence of heat transport (corr=0.33). Thus, it seems that both the convergence of moisture transport and the convergence of heat transport are responsible for the variation of specific humidity and temperature in the lower troposphere. On the other hand, the convergence of horizontal transport of moisture cannot explain one essential element of the specific humidity anomaly—the mean of the anomalous specific humidity. As can be seen in Figure R5a, the mean of moisture convergence is close to $0.6 \times 10^{-6}$ g/kg/sec, which amounts to ~0.05 g/kg of

moisture. This value explains only about 17% of the mean value of anomalous specific humidity (~0.3 g/kg); the remainder should derive from a vertical process.

[Figure]

Figure R5. The daily time series of anomalous specific humidity and anomalous moisture convergence averaged over the sea ice loss region (21°-79.5°E × 75°-79.5°N) in the Barents-Kara Seas. This particular time series is derived from the regressed loading vectors associated with the sea ice loss mode.

10 Consider the following moisture conservation equation:

$$\frac{\partial q}{\partial t} = -\vec{u} \cdot \nabla q + S \doteq -\nabla \cdot (q\vec{u}) + S = -\nabla_h \cdot (q\vec{u}) - \frac{\partial (qw)}{\partial z} + S. \tag{3}$$

According to Fig. R5, the convergence of the horizontal moisture transport is not so effective as the convergence of the vertical moisture transport in the equation above in terms of increasing the mean of specific humidity. A positive convergence is offset by a negative convergence and vice versa, resulting in a small net increase in the mean of specific

15 humidity. As can be seen in Fig. R6, the anomalous evaporation due to sea ice reduction is positive throughout the winter and its magnitude is reasonable in comparison with the increase in specific humidity. The two time series in Fig. R6 are negatively correlated (except for the mean), indicating that increased (decreased) specific humidity due to positive (negative) convergence of moisture transport reduces (augments) evaporation from the surface of the ocean; this is a reasonable explanation according to the bulk formula.

Likewise, the variation of the thermal advection and the subsequent convergence of the heat flux are highly correlated with the variation of downward longwave radiation and the lower tropospheric (850 hPa) temperature (see Fig. R5b). On the other hand, the small mean value of the convergence of the horizontal heat flux cannot explain the significant nonzero mean of the anomalous downward longwave radiation or the anomalous lower tropospheric (850 hPa) temperature. Thus, we

25 conclude that the vertical process should be invoked in order to account for the significant changes in the means of the

[Figure]

Figure R6. The daily variation of specific humidity (red) and evaporation (blue) averaged over the region of sea ice reduction (21°-79.5°E × 75°-79.5°N) in the Barents-Kara Seas.

variables over the Barents-Kara Seas. We did not simply ignore the contributions of moisture transport and heat transport from lower latitudes. Rather, this is a serious issue and requires more detailed calculation and convincing demonstration, which we considered beyond the scope of the present paper. We, however, acknowledge that we restricted ourselves to processes acting in the Arctic, ignoring the forcing from lower latitudes. Based on this discussion we made the following changes: [P3 L4-5: It should be noted that our discussion is restricted to processes in the Arctic; forcing from lower latitudes can also be important in the process of Arctic amplification and sea ice reduction.]

C4: I further do not see any justification for fitting an exponential to the time series of sea ice loss in the Barents-Kara seas, and far less for using that fit to make a prediction on when this ocean area would remain ice-free in winter.

[Figure]

Figure R7. Actual sea ice change in the sea-ice loss region (21°–79.5°E, 75°–79.5 °N) of the Barents and Kara Seas (black dotted curve), sea ice change according to the sea ice loss mode (red curve), projections based on the exponential fitting (blue dashed curve), quadratic fitting (dash-dot curve), and linear fitting (dotted curve) of the PC time series.

R4: As can be seen in Fig. 1h in the manuscript, the exponential fitting looks reasonable in describing the change in sea ice concentration in the Barents-Kara Seas. One can use a linear or quadratic fit to make a similar prediction (see Fig. R7). Predictions are predictions whether it is based on the exponential fitting or the fitting of a low order polynomial; uncertainty is inherent in a prediction. We added the predictions based on linear fit and quadratic fit as supplementary information for the benefit of the readers. The reason why we chose the exponential fit (not on the sea ice concentration but on the PC time series of the sea ice loss mode in Fig. 1g in the manuscript) is that it yields the least residual error. The residual error based on a quadratic fit is similar to that of the exponential fit whereas a linear fit yields the largest residual variance among the three. We included some of this discussion and Fig. R7 as supplementary information. We also modified the text as follows: [P1 L24: … sea ice will completely disappear in the Barents and Kara Seas by as early as 2025, although a conservative linear fit delays it until 2065.] [P7 L31: We fitted an exponential curve to the amplitude time series of the sea ice loss mode (Fig. 1g); an exponential fitting is chosen, since it minimizes the residual error. Our calculation shows that sea ice in the sea-ice loss region (21°–79.5°E, 75°–79.5 °N) of the Barents and Kara Seas may completely melt by as early as 2025 (Fig. 1h) unless impeded by other naturally occurring variability. A quadratic fit results in a similar result (2030 instead of 2025). A linear fit, the most conservative of the three but with the largest residual error, predicts a complete disappearance of sea ice in this area by 2065 (see Fig. S4).] We also updated the sea ice concentration curve using the 2017 sea ice data (see new Fig. 1h).

C5: In conclusion, I regret to say that the manuscript fails to meet basic scientific standards.

R5: We have addressed all specific comments and are not sure what standards we have failed to meet. We would be happy to address those if they would be identified.

**Understanding the Mechanism of Arctic Amplification and Sea Ice Loss**

[revised manuscript text omitted]

$$T(r,t) = \sum_n B_n(r,t)T_n(t), \qquad B_n(r,t) = B_n(r,t+d), \qquad (1)$$

where $B_n(r,t)$ depicts daily winter evolution of the $n$th physical process and $T_n(t)$ describes how the amplitude of the evolution varies on a longer time scale, and $r$ and $t$ denote location and time, respectively. Since the nested period $d = 90$ days, each loading vector, $B_n(r,t)$, consists of 90 spatial patterns which depict evolution of a variable throughout the winter. These winter evolution patterns, $B_n(r,t)$, repeat every winter, but its amplitude varies from one year to another according to the corresponding PC time series. CSEOF loading vectors are mutually orthogonal to each other in space and time and represent distinct physical processes. The principal component (PC) time series, $T_n(t)$ are uncorrelated with (and are often nearly independent of) each other. Each loading vector depicts a temporal evolution of spatial patterns seen in a physical process (such as El Niño or seasonal cycle), and corresponding PC time series describes a long-term modulation of the amplitude of the physical process. Thus, the CSEOF technique is suitable for extracting and depicting temporal evolution of (nearly independent) physical processes and often yields valuable insight that cannot be attained from single spatial pattern.

In order to make suitable physical interpretation of the analysis results, CSEOF analysis is conducted on a number of key variables. It is, then, extremely important to make CSEOF loading vectors derived from individual variables to be physically consistent with each other. For the purpose of generating physically consistent CSEOF loading vectors, regression analysis is carried out in CSEOF space (Kim et al., 2015). A target variable is chosen such that its major CSEOF modes best depict the physical process under investigation; target variable is sea ice concentration in the present study.

Once CSEOF analysis on the "target" variable is completed as in (1), physically consistent loading vectors of another variable, called the "predictor" variable, are obtained as follows:

Jinju Kim 7/7/2017 9:50 AM

Step 1: $P(r,t) = \sum_n C_n(r,t)P_n(t)$   (CSEOF analysis on a new variable)   (2)

Step 2: $T_n(t) = \sum_{m=1}^{M} \alpha_m^{(n)} P_m(t)$   (regression on PC time series)   (3)

Step 3: $Z_n(r,t) = \sum_{m=1}^{M} \alpha_m^{(n)} C_m(r,t)$   (regressed loading vector)   (4)

Then, the target and predictor variables can be written as

$$\{T(r,t), P(r,t)\} = \sum_n \{B_n(r,t), Z_n(r,t)\}T_n(t).$$   (5)

Namely, the loading vectors of the two variables, $B_n(r,t)$ and $Z_n(r,t)$, share an identical PC time series, $T_n(t)$, for each mode. As a result, the evolution of a physical process manifested as $B_n(r,t)$ and $Z_n(r,t)$ in two different variables is governed by a single amplitude time series. Otherwise, $B_n(r,t)$ and $Z_n(r,t)$ do not represent the same physical process and henceforth are not physically consistent. This process can be repeated for other predictor variables. As a result of regression, then, entire data can be written in the form

$$Data(r,t) = \sum_n \{B_n(r,t), Z_n(r,t), U_n(r,t), \dots\}T_n(t) ,$$   (6)

where the terms in curly braces denote physically consistent evolutions derived from various physical variables. A rigorous mathematical explanation of the regression analysis in CSEOF space can be found in Kim et al. (2015).

Aside from the winter seasonal cycle, the first CSEOF mode derived from the daily winter sea ice concentration data in the Arctic depicts sea ice loss and associated Arctic warming in the Barents and Kara Seas. This mode explains 24% of the total variability of the sea ice concentration in the Artic Ocean and is the focus of investigation in the present study.

**3 Results and Discussion**

Figure 1 shows the winter-averaged pattern of $B_1(r,t)$ together with the regressed patterns from other variables (the terms in the curly braces in (6)). We refer to it as the sea ice loss mode, since the loading vector (Fig. 1a; see also Fig. 2) and the amplitude time series (Fig. 1g) describes the sea ice reduction, together with natural variability of sea ice concentration, in the Barents and Kara Seas during the past 37 years (Fig. 1h). The pattern of sea ice reduction (Fig. 1a) is nearly identical with the trend pattern of sea ice concentration in the Arctic Ocean (see Fig. S1 in the supplementary information). As can be seen in Fig. 1h, the sea ice reduction trend in the Barents and Kara Seas (boxed area in Fig. 1a) is faithfully captured by this mode. In particular, the rate of sea ice loss has significantly increased since 2004-2005 (Vihma, 2014). In association with the sea ice reduction, 2 m air temperature, 850 hPa temperature, specific humidity, upward longwave radiation, downward longwave radiation, and upward heat flux have increased significantly over the region of major sea ice reduction (21°-79.5° E × 75°-79.5° N) (black boxed area in Fig. 1a). As can be seen in Figs. 1a, 1c and 1e, the central areas of anomalous 2 m air temperature, upward longwave radiation and turbulent (sensible + latent) heat flux match well with the region of sea ice loss (Screen and Simmonds, 2010b). On the other hand, the centers of downward longwave radiation and specific humidity match well with that of the 850 hPa air temperature (Figs. 1b, 1d, and 1f).

Sea ice concentration varies slightly on a daily basis, and its fluctuation is less than 2% from the mean value of −14.7% throughout the winter (Fig. 2). In accordance with the reduced sea ice concentration, upward longwave radiation

flux is increased from the warmer sea surface exposed to air. Multiplying the amplitude (PC) time series (Fig. 1g) with the loading vector (Fig. 2) of the sea ice loss mode as in (1), actual sea ice concentration time series is obtained as in Fig. 1h. According to Fig. 1h, sea ice concentration has decreased by ~40% during the last 37 years.

Figure 3 shows the anomalous surface (2 m) air temperature, the lower tropospheric geopotential height and wind and the vertical section of anomalous temperature, geopotential height and wind along 60°E and 80°N associated with sea ice reduction. A significant warming is seen in the lower troposphere (e.g., Serreze and Francis, 2006; Serreze et al., 2007; Screen et al., 2013). Note that the anomalous temperature pattern is similar to the second EOF pattern in Graversen et al. (2008). The anomalous temperature and geopotential height are consistent according to the hydrostatic equation (see Fig. S2). Anomalous wind and geopotential height are consistent according to the thermal wind equation. As can be seen, an anticyclonic circulation is established over the region of sea ice loss. This anticyclonic circulation results in advection of warmer air over the Barents and Kara Seas and advection of colder air over the mid-latitude East Asia (Kim and Son, 2016).

The winter-averaged patterns of anomalous downward longwave radiation and specific humidity look fairly similar to that of 850 hPa air temperature (Figs. 4a and 4b). It appears that the increased downward longwave radiation is the result of the tropospheric warming (Fig. 3). Specific humidity also increases with the tropospheric warming. Note specifically that these changes are observed over or close to the region of sea ice reduction. The patterns of total cloud liquid water and total cloud ice water, which are the key variables for the formation of clouds, also exhibit a strong response over the region of sea ice reduction although their centers of action are shifted toward the Greenland Sea (Fig. 4d). The pattern of total cloud cover, however, does not show any strong cloud activity over the region of sea ice reduction (Fig. S3 in the supplementary information); it should be understood that cloud cover is a difficult variable to simulate accurately in a reanalysis model. Therefore, we postulate that the increased downward longwave radiation is due to the increased 850 hPa air temperature and the greenhouse effect produced by the increased specific humidity. Further note that net (upward minus downward) longwave radiation is positive over the region of major sea ice reduction, whereas it is slightly negative over the surrounding areas (Fig. 4c). Thus, at the surface level, there is a net loss of longwave energy over the region of sea ice reduction, while there is a net gain of longwave radiation over the surrounding area.

A prominent source of energy available for heating the atmospheric column is the increased turbulent heat flux from the sea surface exposed to air due to sea ice reduction (Fig. 5). Figure 6 shows the winter daily variations of the regressed loading vectors in (6) (terms in curly braces) averaged over the region of sea ice reduction (21°-79.5° E × 75°-79.5° N); it may be interpreted as atmospheric response to the sea ice reduction shown in Fig. 2. Although the total (area-weighted) magnitudes of sensible and latent heat fluxes are generally smaller than those of upward and downward longwave radiation (see Fig. 6a), turbulent heat flux (see Fig. 5) is locally more pronounced than longwave radiations (Deser et al., 2010). Furthermore, the combined effect of turbulent heat flux is about 6 times larger than that of longwave radiation, since upward and downward longwave radiation tends to offset each other and the resulting net longwave radiation is comparatively smaller than the net upward turbulent heat flux (Fig. 6a). In the presence of turbulent heat flux, air temperature and, henceforth, downward longwave radiation can increase continually leading to further sea ice reduction.

Jinju Kim 7/7/2017 9:54 AM

Jinju Kim 7/7/2017 10:21 AM

Jinju Kim 7/7/2017 9:55 AM

Jinju Kim 7/7/2017 9:56 AM

[revised manuscript text omitted]

of 2025). A linear fit, the most conservative of the three but with the largest residual error, predicts a complete disappearance of sea ice in this area by 2065 (see Fig. S4).

It should be pointed out that this feedback process could develop in other areas of the Arctic Ocean. If sea ice refreezing is delayed in late fall/winter, increased turbulent heat flux from the open sea surface will make it more difficult for

5 sea surface to refreeze, ultimately leading to the feedback process in Fig. 8. It is, of course, difficult to determine when this should occur, since environmental factors differ from one location to another. Finally, it should be mentioned that the feedback process does not seem to be sensitive to a choice of the dataset. A similar experiment conducted by using the NCEP reanalysis data produces essentially identical results except for a slight overestimation of the strength of the anomalous patterns in Fig. 1a-f and Fig. 6 (see Figs. S5 and S6 in the supplementary information).

[revised manuscript text omitted]
 red curve in Fig. 1h is obtained by multiplying the loading vector of sea ice concentration (Fig. 1a) averaged in the boxed area with the amplitude time series (Fig. 1g) according to (1). The green contours in (b)-(f) represent sea ice concentration in (a). The numbers in parenthesis are contour intervals and negative contours are dashed.

**Figure 2.** Anomalous daily sea ice concentration (blue) and upward longwave radiation averaged over the region of sea ice loss (21°-79.5° E × 75°-79.5° N) with respective mean values (straight lines). Winter days are counted from December 1.

**Figure 3**. Winter-averaged patterns of anomalous atmospheric condition: (a) 2m air temperature, (b) lower tropospheric (1000-900 hPa) geopotential height and wind, (c) vertical cross section along 60° E of lower tropospheric (1000-850 hPa) air temperature, geopotential height and wind, and (d) along 80°N. Contour intervals are in parenthesis in (a) and (b). Temperature is in shading (0.4 K), geopotential height is in black contours (3 m), and (c) zonal and (d) meridional winds are in blue contours (0.2 m s$^{-1}$).

**Figure 4**. Winter-averaged patterns of (a) 850 hPa air temperature (shading) and 2 m air temperature (contour), (b) 900-hPa specific humidity (shade) and downward longwave radiation at surface (contour), (c) net (upward minus downward) longwave radiation at surface (shade) and SAT (contour), and (d) total cloud liquid water (shade) and total cloud ice water (contour) for the sea ice loss mode. The red contour is drawn at the value of the contour interval. The green contours in (a)-(d) represent the reduction of sea ice concentration.

**Figure 5**. Winter average pattern of sea ice loss mode in the Barents and Kara Seas: (a) sea ice (%, shading), 2 m air temperature (red contour) and 850 hPa temperature (black contour), (b) upward longwave radiation (red contour) and downward longwave radiation (black contour), (c) sensible heat flux (red contour) and latent heat flux (black contour), and (d) net energy balance (sensible heat flux + latent heat flux + upward longwave radiation – downward longwave radiation).

**Figure 6**. Daily patterns of variability over the region of sea ice loss (21°-79.5° E × 75°-79.5° N): (a) upward longwave radiation (blue dashed), downward longwave radiation (blue dotted), net longwave radiation (blue solid) with its mean value (blue straight line), sensible heat flux (red dashed), latent heat flux (red dotted), and turbulent heat flux (red solid) with its mean value (red straight line), (b) 2 m air temperature (red), 850 hPa air temperature × 2 (black), and upward longwave radiation (blue), and (c) same as (b) except for the regressed downward longwave radiation (blue). The straight lines in (b) and (c) represent the winter mean value of anomalous 2 m air temperature. Correlation of upward and downward longwave radiations with 2 m air temperature is respectively 0.88 and 0.91, whereas with 850 hPa air temperature is 0.66 and 0.85.
* * *
Jinju Kim 7/7/2017 10:25 AM

Jinju Kim 7/7/2017 10:01 AM

Jinju Kim 7/7/2017 10:02 AM

Jinju Kim 7/7/2017 10:02 AM

Jinju Kim 7/7/2017 10:24 AM

**Figure 7.** Lagged correlations: (a) correlation of upward (solid lines) and downward (dotted lines) longwave radiations with 2 m air temperature (blue), 850 hPa temperature (red), and sea ice concentration (black), and (b) a blow-up of the boxed region in (a). Longwave radiation lags the other variable for a positive lag. Lagged correlation between 2 m air temperature and 850 hPa air temperature (black dashed line); 2 m air temperature leads 850 hPa temperature for a positive lag.

5 **Figure 8**. A proposed mechanism of polar amplification. Increased net upward energy flux increases air temperature. As a result, downward longwave radiation increases, which results in sea ice melting. This loop seems to amplify by ~8.9% annually.

Jinju Kim 7/7/2017 10:03 AM

(a) SIC (2 %) & 2m AIR T (0.5° C)    (b) 1000-850 hPa SH (3×10⁻² g Kg⁻¹)

(c) ULW at SFC (2 W m⁻²)    (d) DLW at SFC (2 W m⁻²)

(e) TURBULENT FLUX (4 W m⁻²)    (f) 850 hPa T (0.2° C)

[Figure]

Jinju Kim 7/7/2017 10:07 AM

(a) SIC (2 %) & 2m AIR T (0.5° C)

(c) ULW at SFC (2 W m⁻²)

(e) TURBULENT FLUX (4 W m⁻²)

[Figure]

[Figure]

**Figure 1.** Winter (Dec. 1-Feb. 28) average patterns of sea ice loss mode: (a) sea ice (shading) and 2 m air temperature (contour), (b) 1000-850 hPa specific humidity, (c) upward longwave radiation, (d) downward longwave radiation, (e) turbulent (sensible + latent) heat flux, (f) 850 hPa air temperature, (g) the corresponding amplitude change (red solid curve) and the amplification curve (blue dashed curve), and (h) actual sea ice change in the sea-ice loss region (21°–79.5° E × 75°–79.5° N; the boxed area in (a)) of the Barents and Kara Seas (black dotted curve; extended until 2017 based on new data), sea ice change according to the sea ice loss mode (red curve), projection based on the amplification curve (blue dashed curve). The red curve in Fig. 1h is obtained by multiplying the loading vector of sea ice concentration (Fig. 1a) averaged in the boxed area with the amplitude time series (Fig. 1g) according to (1). The green contours in (b)-(f) represent sea ice concentration in (a). The numbers in parenthesis are contour intervals and negative contours are dashed.

Jinju Kim 7/7/2017 10:25 AM

[Figure]

**Figure 2.** Anomalous daily sea ice concentration (blue) and upward longwave radiation averaged over the region of sea ice loss (21°-79.5° E × 75°-79.5° N) with respective mean values (straight lines). Winter days are counted from December 1.

[Figure]

Jinju Kim 7/7/2017 10:09 AM

Jinju Kim 7/7/2017 10:10 AM

[Figure]

[Figure]

Jinju Kim 7/7/2017 10:10 AM

Jinju Kim 7/7/2017 10:11 AM

**Figure 3.** Winter-averaged patterns of anomalous atmospheric condition: (a) 2 m air temperature, (b) lower tropospheric (1000-900 hPa) geopotential height and wind, (c) vertical cross section along 60° E of lower tropospheric (1000-850 hPa) air temperature, geopotential height and wind, and (d) along 80° N. Contour intervals are in parenthesis in (a) and (b). Temperature is in shading (0.4 K), geopotential height is in black contours (3 m), and (c) zonal and (d) meridional winds are in blue contours (0.2 m s$^{-1}$).

[Figure]

[Figure]

**Figure 4**. Winter-averaged patterns of (a) 850 hPa air temperature (shading) and 2 m air temperature (contour), (b) 900-hPa specific humidity (shade) and downward longwave radiation at surface (contour), (c) net (upward minus downward) longwave radiation at surface (shade) and SAT (contour), and (d) total cloud liquid water (shade) and total cloud ice water (contour) for the sea ice loss mode. The red contour is drawn at the value of the contour interval. The green contours in (a)-(d) represent the reduction of sea ice concentration.

Jinju Kim 7/7/2017 10:11 AM

Jinju Kim 7/7/2017 10:12 AM

Jinju Kim 7/7/2017 10:26 AM

[Figure]

[Figure]

Jinju Kim 7/7/2017 10:12 AM

**Figure 5**. Winter average pattern of sea ice loss mode in the Barents and Kara Seas: (a) sea ice (%, shading), 2 m air temperature (red contour) and 850 hPa temperature (black contour), (b) upward longwave radiation (red contour) and downward longwave radiation (black contour), (c) sensible heat flux (red contour) and latent heat flux (black contour), and (d) net energy balance (sensible heat flux + latent heat flux + upward longwave radiation – downward longwave radiation).

[Figure]

**Figure 6.** Daily patterns of variability over the region of sea ice loss (21°-79.5° E × 75°-79.5° N): (a) upward longwave radiation (blue dashed), downward longwave radiation (blue dotted), net longwave radiation (blue solid) with its mean value (blue straight line), sensible heat flux (red dashed), latent heat flux (red dotted), and turbulent heat flux (red solid) with its mean value (red straight line), (b) 2 m air temperature (red), 850 hPa air temperature × 2 (black), and upward longwave radiation (blue), and (c) same as (b) except for the regressed downward longwave radiation (blue). The straight lines in (b) and (c) represent the winter mean value of anomalous 2 m air temperature. Correlation of upward and downward longwave radiations with 2 m air temperature is respectively 0.88 and 0.91, whereas with 850 hPa air temperature is 0.66 and 0.85.

[Figure]

[Figure]

**Figure 7**. Lagged correlations: (a) correlation of upward (solid lines) and downward (dotted lines) longwave radiations with 2 m air temperature (blue), 850 hPa temperature (red), and sea ice concentration (black), and (b) a blowup of the boxed region in (a). Longwave radiation lags the other variable for a positive lag. Lagged correlation between 2 m air temperature and 850 hPa air temperature (black dashed line); 2 m air temperature leads 850 hPa temperature for a positive lag.

[Figure]

Increased $T$
(0.09 K)

Increased $LW^{\downarrow}$
(0.91 W m$^{-2}$)

Increased $FL^{\uparrow}$
(1.02 W m$^{-2}$)

Sea ice reduction
(1%)

**Figure 8.** A proposed mechanism of polar amplification. Increased net upward energy flux increases air temperature. As a result, downward longwave radiation increases, which results in sea ice melting. This loop seems to amplify by ~8.9 % annually.

**Supplementary Information**

**Understanding the Mechanism of Arctic Amplification and Sea Ice Loss**

Kwang-Yul Kim[1], Jinju Kim[1], Saerim Yeo[2], Hanna Na[3], Benjamin D. Hamlington[4], and Robert R. Leben[5]

[1]School of Earth and Environmental Sciences, Seoul National University, 1 Gwanak-ro, Gwanak-gu, Seoul 08826, Republic of Korea
[2]APEC Climate Center 1463, Haeundae-gu, Busan 48058, Republic of Korea
[3]Ocean Circulation and Climate Research Center, Korea Institute of Ocean Science and Technology, Ansan, 15627, Republic of Korea
[4]Department of Ocean, Earth and Atmospheric Sciences, 4600 Elkhorn Avenue, Room 406, Old Dominion University, Norfolk, Virginia 23529, USA
[5]Colorado Center for Astrodynamics Research, Department of Aerospace Engineering Sciences, ECNT 320, 431 UCB, University of Colorado, Boulder, Colorado 80309-0431, USA

*Correspondence to*: Kwang-Yul Kim (kwang56@snu.ac.kr)

[Figure]

**Figure S1**. (a) The yearly trend (%) of sea ice reduction in the Arctic Ocean during 1979-2016, (b) the winter averaged loading vector of the sea ice loss mode, (c) the corresponding PC (amplitude) time series, and (d) actual sea ice concentration in the boxed area (black curve), sea ice concentration according to the sea ice loss mode (red curve) and a projection (red dashed curve) based on the exponential fit of the amplitude time series in (c).

Figure S1a is obtained based on a linear trend of sea ice concentration at each grid point based on the ERA-Interim sea ice concentration from 1979-2016. This pattern is nearly identical with the sea ice loss mode discussed in the main text (Fig. S1b). Sea ice reduction is most conspicuous in the Barents-Kara Seas. The amount of sea ice reduction based on the sea ice loss mode is obtained by multiplying the loading vector (Fig. S1b) with its amplitude time series (Fig. S1c), resulting in Fig. S1d. As can be seen in Fig. S1d, the sea ice concentration change due to the sea ice loss mode (red curve) is similar to the actual data (black curve) with a fairly similar rate of trend.

[Figure]

**Figure S2.** The pressure layer thickness ($\Delta Z = Z(p_1) - Z(p_0)$) derived from the geopotential height pattern in Fig. 3 in the text and that derived from the hydrostatic equation (contour). The red contour represents the thickness of 1.5 m. The level
5   $p_1$ is the level used for plotting and $p_0$ is the pressure level below $p_1$ at the interval of 25 hPa.

The shaded geopotential height anomaly in this figure is obtained directly from the geopotential height field in Fig. 3 in the main text, i.e.,

10        $$(dZ)_j = Z_j - Z_{j-1}, \tag{1}$$

where $j$ is an index for the vertical level. The contoured geopotential height anomaly is obtained from the temperature field in Fig. 3 in the text, i.e.,

$$(dZ)_j = -\frac{R\langle T \rangle_j}{g}(d \ln p)_j, \tag{2}$$

where

15        $$(dZ)_j = Z_j - Z_{j-1}, \quad \langle T \rangle_j = (T_j + T_{j-1})/2, \quad (d \ln p)_j = \ln p_j - \ln p_{j-1}. \tag{3}$$

As can be seen in the figure, the anomalous geopotential height field is nearly in hydrostatic balance with the anomalous temperature field. The difference is partially due the use of layer mean temperature $\langle T \rangle$ in a finite-difference approximation of the hydrostatic equation in (2). Thus, it seems that the release of energy in the form of radiation and heat flux changes the temperature, and geopotential height in the lower troposphere adjusts in accordance with the hydrostatic balance.

[Figure]

(a) 850 hPa T (0.2° C) & SAT (0.5° C)

(b) TCC (0.5 %)

(c) DLW & ULW at SFC (2 W m⁻²)

(d) SH (0.02 g kg⁻¹) & DLW (2 W m⁻²)

**Figure S3**.  The DJF patterns of 850 hPa air temperature (shading) and 2 m air temperature (contour) (a), total cloud cover (b), downward (shade) and upward (contour) longwave radiation at surface (c), and 900-hPa specific humidity (shade) and downward longwave radiation at surface (contour) (d) for the sea ice loss mode.  The green and purple contours in (a)-(d) represent the reduction of sea ice concentration.

This figure shows the winter (DJF) averaged patterns of several key variables associated with the sea ice reduction.  As can be seen, anomalous patterns of all the variables exhibit strong coherence with that of sea ice reduction except for total cloud cover.  The pattern of total cloud cover associated with the sea ice loss mode does not exhibit any strong cloud activity over the region of sea ice reduction, suggesting little connectivity between sea ice reduction and change in cloud cover.  However, total cloud liquid water and total cloud ice water, which are two important elements for the production of clouds are reasonably consistent with the pattern of sea ice reduction (see Fig. 4 in the main text).

[Figure]

**Figure S4**. Actual sea ice change in the sea-ice loss region (21°–79.5°E, 75°–79.5 °N) of the Barents and Kara Seas (black dotted curve; updated until Feb. 2017 using new dataset), sea ice change according to the sea ice loss mode (red curve), projections based on the exponential fitting (blue dashed curve), quadratic fitting (dash-dot curve), and linear fitting (dotted curve) of the PC time series.

This figure shows projections of sea ice concentration in the sea-ice loss region of the Barent-Kara Seas based on a linear fit (dotted curve), a quadratic fit (dash-dot curve), and an exponential fit (dashed). Residual variance is measured by

$$\varepsilon^2 = var\big(S(t) - F(t)\big),$$

where $S(t)$ is the sea ice concentration curve (black curve in Fig. S4) and $F(t)$ is a fit. The exponential fit results in the least residual variance, whereas the linear fit the largest residual variance. The residual variance of the quadratic fit is similar to that of the exponential fit. Sea ice in the region (21°–79.5°E, 75°–79.5 °N) disappears completely by 2025 (2030, 2065) according to the exponential (quadratic, linear) fit. According to the newly available data, sea ice concentration in this area is the lowest during 2016 winter (Dec. 2016-Feb. 2017; see the dotted line in Fig. S4).

[Figure]

**Figure S5**. Winter (Dec. 1-Feb. 28) average patterns of sea ice loss mode derived from the NCEP reanalysis data: (a) sea ice (shading) and 2 m air temperature (contour), (b) 1000-850 hPa specific humidity, (c) upward longwave radiation, (d) downward longwave radiation, (e) turbulent (sensible + latent) heat flux, (f) 850 hPa air temperature.

This figure shows the winter-averaged patterns of key variables from the NCEP reanalysis product (1979-2016) associated with the sea ice loss mode. The target variable is the ERA-Interim sea ice concentration as in the main text. This figure is fairly similar to Fig. 1a-f in the text except for a small difference in magnitude. Thus, an essentially identical physical

5    process is identified in the NCEP reanalysis product. This result indicates that the physical mechanism addressed in the present study is not overly sensitive to the choice of the dataset for analysis.

[Figure]

**Figure S6**. Daily pattern of variability over the region of sea ice loss (21°-79.5°E × 75°-79.5°N) derived from the NCEP reanalysis data: (a) 2 m air temperature (red), 850 hPa air temperature × 2 (black), and upward longwave radiation (blue), and (b) same as (a) except for the regressed downward longwave radiation (blue). The straight line represents the winter mean value of anomalous 2 m air temperature. Correlation of upward and downward longwave radiation with 2 m air temperature is respectively 0.95 and 0.94, whereas correlation with 850 hPa air temperature is respectively 0.81 and 0.86.

This figure shows the daily evolution of surface (2 m) air temperature, 850 hPa air temperature, upward longwave radiation and downward longwave radiation during winter in response to sea ice reduction in the Barents-Kara Seas as in Fig. 2 in the main text. This result is obtained by using the NCEP reanalysis data. A comparison with Fig. 6 in the main text shows that the response of atmospheric variables to the sea ice reduction in the Barents-Kara Seas as identified from the NCEP reanalysis product is fairly similar to that derived from the ERA-Interim reanalysis product. This figure together with Fig. S4 indicates that the physical process of sea ice reduction and Arctic warming discussed in the text is not sensitive to the choice of analysis dataset.